# IL-7-dependent compositional changes within the γδ T cell pool in lymph nodes during ageing lead to an unbalanced anti-tumour response

Hung-Chang Chen[1] (iD), Nils Eling[1,2,†] (iD), Celia Pilar Martinez-Jimenez[1,3,4,†] (iD), Louise McNeill O'Brien[1], Valentina Carbonaro[1] (iD), John C Marioni[1,2,3] (iD), Duncan T Odom[1,3,5] (iD) & Maike de la Roche[1,*] (iD)

## Abstract

How the age-associated decline of immune function leads to increased cancer incidence is poorly understood. Here, we have characterised the cellular composition of the γδ T-cell pool in peripheral lymph nodes (pLNs) upon ageing. We find that ageing has minimal cell-intrinsic effects on function and global gene expression of γδ T cells, and γδTCR diversity remains stable. However, ageing alters TCRδ chain usage and clonal structure of γδ T-cell subsets. Importantly, IL-17-producing γδ17 T cells dominate the γδ T-cell pool of aged mice—mainly due to the selective expansion of Vγ6[+] γδ17 T cells and augmented γδ17 polarisation of Vγ4[+] T cells. Expansion of the γδ17 T-cell compartment is mediated by increased IL-7 expression in the T-cell zone of old mice. In a Lewis lung cancer model, pro-tumourigenic Vγ6[+] γδ17 T cells are exclusively activated in the tumour-draining LN and their infiltration into the tumour correlates with increased tumour size in aged mice. Thus, upon ageing, substantial compositional changes in γδ T-cell pool in the pLN lead to an unbalanced γδ T-cell response in the tumour that is associated with accelerated tumour growth.

**Keywords** ageing; IL-7; lymph node; tumour response; γδ T-cell lineage
**Subject Categories** Ageing; Cancer; Immunology

See also: **I Prinz & I Sandrock** (August 2019)

## Introduction

A decline of potent T-cell responses during ageing has been linked to increased susceptibility to infection and the drastic rise in cancer incidence observed in elderly mice and humans [1–3]. Three interrelated components of the immune response are affected by immunosenescence: the immune cells themselves, the supporting lymphoid organs and circulating factors that guide responses of immune cells as well as lymphoid organs [2]. In αβ T cells, a restricted TCR repertoire, loss of intrinsic cell functions, compromised priming and chronic and low-grade inflammation have been associated with impaired anti-tumour responses [1].

γδ T cells are unconventional T cells that combine adaptive features with rapid innate-like functions to mediate responses to infection, tissue damage and cancer [4]. In contrast to αβ T cells that acquire cytokine-secreting effector functions upon activation in the periphery, murine γδ T cells acquire their effector potential in the thymus where they differentiate into either IFN-γ-producing (γδ1) or IL-17-producing (γδ17) lineages [5]. It is this pre-activated differentiation state and unique innate-like activities that enable γδ T cells to rapidly infiltrate into inflammatory sites, such as tumours, in the periphery. Here, they modulate the early local microenvironment and subsequent αβ T-cell responses by secretion of pro-inflammatory cytokines [6–8].

The anti-tumour effects of γδ T cells are well-established in various cancer models—mainly due to their extensive cytotoxic capacity and IFN-γ production [9,10]. However, tumour-promoting roles of the IL-17-producing γδ T-cell subsets have emerged [11,12]. Pro-tumour mechanisms of IL-17 produced by Vγ4[+] and Vγ6[+] γδ T cells include the promotion of angiogenesis [13] and recruitment of immunosuppressive cells, such as myeloid-derived suppressor cells (MDSCs) [14,15] and peritoneal macrophages [16].

Most studies on the functions of γδ T cells have focused on their roles in barrier tissues—mainly skin [17] and gut [18]—and in the tumour mass itself [10]. However, whether γδ T cells, resident in peripheral lymph node (pLN), are important for tumour-specific responses, as they have recently been shown to be for the response to inflammatory stimuli [19–22], remains unclear.

Also, currently unknown is how the γδ T-cell pool in peripheral lymphoid tissues changes upon ageing, and how age-related

1 Cancer Research UK Cambridge Institute, University of Cambridge, Cambridge, UK
2 European Molecular Biology Laboratory, European Bioinformatics Institute (EMBL-EBI), Wellcome Genome Campus, Cambridge, UK
3 Wellcome Sanger Institute, Wellcome Genome Campus, Cambridge, UK
4 Helmholtz Pioneer Campus, Helmholtz Zentrum München, Neuherberg, Germany
5 Division of Signalling and Functional Genomics, German Cancer Research Center (DKFZ), Heidelberg, Germany
*Corresponding author. Tel: +44 1223 769772; E-mail: maike.delaroche@cruk.cam.ac.uk
†These authors contributed equally to this work

alterations may affect the tumour microenvironment. For the first time, we have characterised the γδ T-cell compartment in pLNs during ageing and investigated the functional relevance for regulating anti-tumour immune responses.

We find that, upon ageing, the γδ T-cell pool in pLNs becomes entirely biased towards the γδ17 lineage, while the number of γδ1 T cells is significantly reduced. We establish that this striking γδ17 bias is due to a substantial accumulation of Vγ6$^+$ γδ17 T cells and, in part, an increased γδ17 polarisation of Vγ4$^+$ and Vγ2/3/7 T-cell subsets in old mice. γδ17 lineage expansion is mediated by IL-7, and increased IL-7 production in the pLNs of old mice provides a selective niche for the expansion of γδ17 T cells. Interestingly, γδTCR diversity is not affected, but TCRδ chain usage and clonal substructure are altered upon ageing. Upon tumour challenge, Vγ6$^+$ γδ17 T cells become activated in pLNs, migrate into the tumour and create a pro-tumour microenvironment that is associated with enhanced tumour growth.

These results demonstrate that the γδ T-cell pool in pLNs is essential for shaping the balance of pro- and anti-tumour immune responses. Bias towards the pro-tumorigenic γδ17 lineage during ageing thus may be a crucial contributor to the age-related increase in tumour incidence.

## Results

### γδ17 T cells constitute the majority of the γδ T-cell pool in peripheral lymph nodes of old mice

To determine the effect of ageing on size and composition of the γδ T-cell pool, we analysed inguinal and axillary lymph nodes (here termed peripheral lymph nodes, pLNs) from young (3 months old) and old (> 21 months old) C57BL/6 mice. Upon ageing, the proportion of γδ T cells amongst all CD3$^+$ T cells in pLNs was increased 2-fold (Fig 1A). The absolute number of γδ T cells in pLNs was significantly decreased (Fig 1B) as a consequence of the smaller pLN size in old animals. The maturation status, assessed by the characteristic lack of CD24 expression by mature γδ T cells, was slightly higher in old mice (Appendix Fig S1A). Thus, mature γδ T cells are enriched in the pLNs of old mice.

In the thymus, commitment of γδ T cells towards γδ1 and γδ17 lineages can be distinguished by the expression of CD44 and CD45RB [23]. We first confirmed that this phenotypic segregation of γδ1 (CD44$^+$ CD45RB$^+$) and γδ17 (CD44$^{hi}$ CD45RB$^{neg}$) T cells is also observed in pLNs (Appendix Fig S1B). Upon stimulation with PMA/Ionomycin, CD44$^{hi}$ CD45RB$^{neg}$ cells produce IL-17 but not IFN-γ, whereas CD44$^+$ CD45RB$^+$ cells produce IFN-γ and not IL-17. CD44$^{neg}$ CD45RB$^+$ cells—an intermediate cell population undergoing differentiation towards the γδ1 lineage—produce only limited IFN-γ upon stimulation, and CD44$^{neg}$ CD45RB$^{neg}$ progenitor cells produce neither IL-17 nor IFN-γ. Consistent with previous reports [5,11,23–27], CD44$^{hi}$ CD45RB$^{neg}$ γδ17 T cells were IL-7R$^{hi}$ CCR6$^+$ IL-23R$^{hi}$ CD27$^{neg}$ CD62L$^{neg}$, while CD44$^+$ CD45RB$^+$ γδ1 T cells showed an IL-7R$^{lo}$ CCR6$^{neg}$ IL-23R$^{lo}$ CD27$^{hi}$ CD62L$^{hi}$ phenotype (Appendix Fig S1C). We next defined the contribution of γδ1 and γδ17 lineages to the γδ T-cell pool in pLNs. We found that γδ1 T cells and γδ1 precursor (γδ1$^{int}$) cells constitute > 80% of the γδ T-cell population in pLNs of young mice, whereas γδ17 T cells

represented only 15% (Fig 1C). Strikingly, this bias was reversed in old mice: γδ1 T cells are diminished, and the γδ17 T-cell population increases to 60–80% of total γδ T cells (Fig 1C). pLNs from middle-aged (12 months old) animals showed an intermediate phenotype, suggesting that loss of γδ1 and gain of γδ17 T cells occur gradually upon ageing (Fig EV1A). We further confirmed the age-specific γδ1/γδ17 lineage redistribution using CD27 as an additional marker to separate γδ1 (CD27$^+$) and γδ17 (CD27$^{neg}$) T cells, again observing increased proportion of γδ17 T cells (CD27$^{neg}$) in pLNs of aged mice [28] (Fig EV1B). γδ17 T cells resembled highly activated T cells (CD44$^{hi}$ CD62L$^{neg}$), as previously reported [25]. Interestingly, γδ1 T cells had a central memory-like phenotype (CD44$^{int}$ CD62L$^+$) and γδ1$^{int}$ T cells showed a naïve-like phenotype (CD44$^{neg}$ CD62L$^+$; Fig EV1C and D). Taken together, upon ageing the γδ T-cell population undergoes a dramatic redistribution favouring the γδ17 T-cell lineage.

High-fat diet leading to obesity can result in an increase in γδ17 T cells in the periphery [29,30]. The mice analysed in this study were fed a standard diet, but some old mice were obese. Importantly, both obese and lean old mice presented with a γδ17 bias and we observed no correlation between obesity and the γδ17 phenotype (Fig EV1E), as had been previously seen [29]. Moreover, our analyses of lean, middle-aged (12 months old) mice that displayed an intermediate γδ17 phenotype in pLNs point towards a gradual accumulation of the phenotype with age independent of obesity.

To determine the functional consequence of increased γδ17 T cells in pLNs of old mice, we assessed cytokine production upon in vitro stimulation with PMA/Ionomycin. Overall, the proportion of IL-17-producing CD3$^+$ T cells was increased 6-fold in pLNs from old mice (Fig EV1F). While on average 10% of γδ T cells from young mice produced IL-17, the proportion of IL-17-producing γδ T cells increased to 50% in old mice. In contrast, over 20% of γδ T cells produced IFN-γ in young mice, and this decreased to below 10% of γδ T cells in old mice (Fig 1D). The absolute levels of IL-17 and IFN-γ production by individual activated cells were similar between young and old γδ T cells (Fig EV1G), indicating that, once activated, the cytokine production capacity of γδ T cells is maintained during ageing. Despite γδ T cells representing only 1–2% of total T lymphocytes in pLNs, they constituted approximately half of the IL-17-producing cells upon stimulation (Fig 1E). Memory CD4$^+$ T cells accounted for the remaining IL-17 production in the pLN. However, only half of the old mice showed an increase in IL-17$^+$ memory CD4$^+$ T cells (Fig EV1H), making the increase in γδ17 T cells, the primary cause of the greatly increased IL-17 production in pLNs of old mice. Thus, we conclude that the prevalent IFN-γ response by γδ T cells in young mice becomes skewed towards an IL-17-dominated response during ageing.

### Composition of γδ T-cell subsets in the pLN pool changes during ageing

Based on their TCRγ chain usage, γδ T cells can be classified into different subsets, each with distinct tissue distribution and degree of plasticity with regard to differentiation towards the γδ1 and γδ17 lineage during thymic development or in the periphery (Fig 2A) [5,31]. We sought to uncover the nature of the γδ17 bias observed in pLNs of old mice. Using the strategy described in Fig 2B, we discriminated γδ T-cell subsets (Heilig and Tonegawa nomenclature) [32] according to their lineage commitment. Consistent with

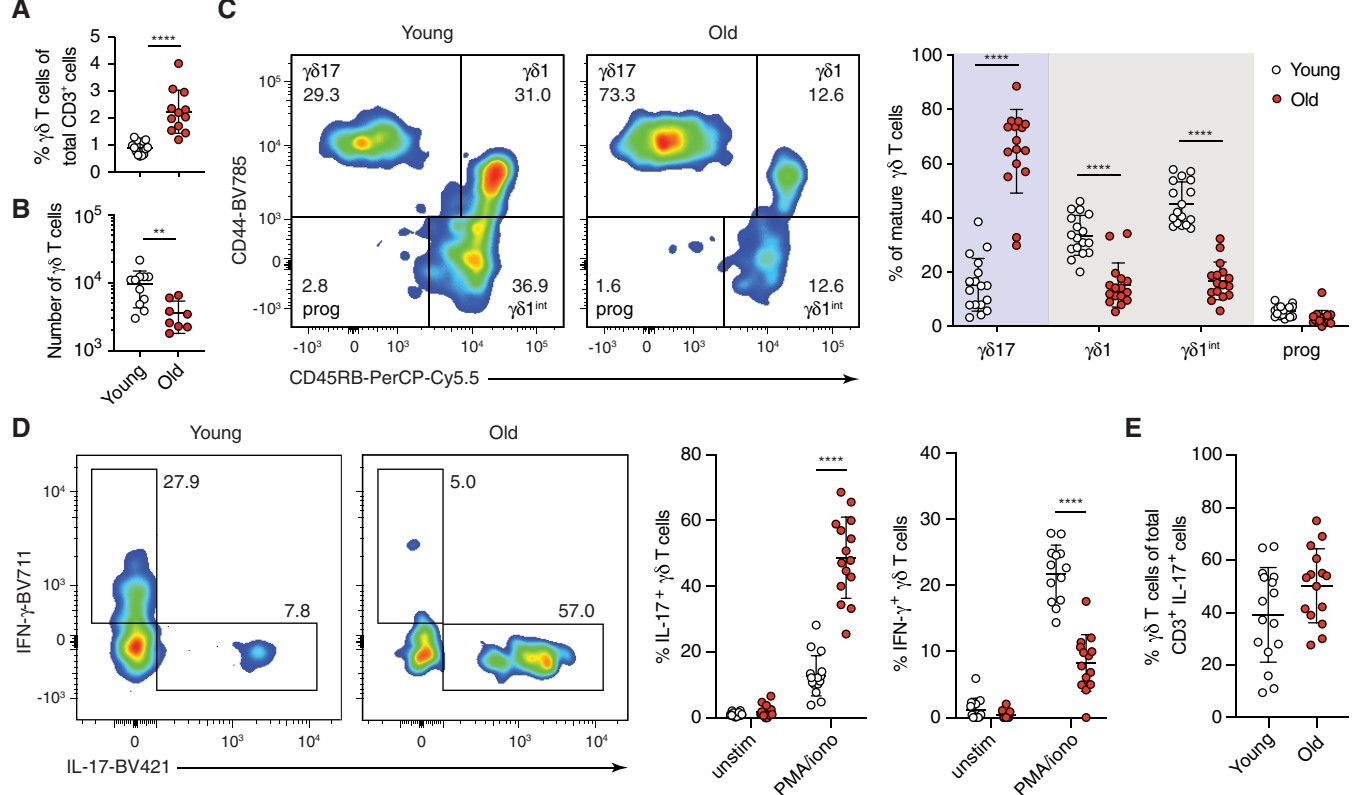

**Figure 1.  γδ T cells from peripheral lymph nodes (pLNs) of old mice are predominantly γδ17-committed.**

Inguinal and axillary LNs were isolated from young (3 months, white circles) and old (> 21 months, red circles) mice.

A   Proportion of γδ T cells in total CD3+ T lymphocytes. Results are from 13 young and 12 old mice (*n* = 5 experiments).

B   Absolute number of γδ T cells from pLNs in young and old mice. Results are from 11 young and 10 old mice (*n* = 4 experiments).

C   CD45RB and CD44 expression of mature (CD24neg) γδ T cells in pLNs of young and old mice. Left: representative FACS plots from 11 independent experiments. Right: percentage of mature γδ17-committed (CD45RBneg CD44hi), γδ1-committed (CD45RB+ CD44+), γδ1-intermediate (CD45RB+ CD44neg) and progenitor (CD45RBneg CD44neg) γδ T cells in pLNs of young and old mice. Results shown are from nine independent experiments with 17 young and 16 old mice.

D   Cell suspensions from pLNs were stimulated with PMA/Ionomycin for 4 h and examined for their production of IL-17 and IFN-γ. Representative FACS plots are gated on CD24neg γδ T cells. Results are from 16 young and 15 old mice (*n* = 6 experiments).

E   Proportion of γδ T cells in total IL-17-producing CD3+ T lymphocytes upon PMA/Ionomycin stimulation. Results are from 16 young and 15 old mice (*n* = 6 experiments).

Data information: Statistical significances for changes in cell proportions were assessed by Mann–Whitney test (A and B), two-way ANOVA (C and D) or unpaired *t*-test (E). Error bars represent SD. **\**P* < 0.01; \*\*\*\**P* < 0.0001.

previous reports [11,31], Vγ1+ and Vγ4+ T cells were the major γδ T-cell subsets in pLNs of young mice (Fig 2C). By contrast, in pLNs of old mice, the Vγ1+ T-cell pool contracted 2-fold, and strikingly the Vγ6+ T-cell pool, which was barely detectable in young mice, expanded more than 10-fold. The Vγ4+ T-cell pool was also slightly smaller in pLNs of old mice (Fig 2C).

Vγ1+ T cells were predominantly committed to the γδ1 lineage in young and old mice, whereas Vγ2/3/7 and Vγ4+ T cells gave rise to both γδ1 and γδ17 T cells (Fig 2D). Although γδ1 T cells constitute the majority of the Vγ2/3/7 and Vγ4+ T-cell pool in the pLNs of young mice, γδ17 T cells were considerably enriched in the Vγ2/3/7 and Vγ4+ T-cell pool in pLNs of old mice (Fig 2D). Vγ6+ T cells are invariant and exclusively committed to the γδ17 lineage in both young and old mice (Fig 2D). Thus, enrichment of γδ17 lineage-committed Vγ6+ T cells and changes in lineage commitment of Vγ4+ and Vγ2/3/7 T cells underpin the increase in γδ17 T cells in pLNs during ageing.

Recently, the local microbiome has emerged to play an important role in the homeostasis of Vγ6+ T cells [33,34]. To control for the possibility that a unique microbiome in our animal facility affects Vγ6+ T-cell homeostasis in the ageing cohort, we analysed pLNs from young and old mice housed in a different animal facility and obtained identical results (Appendix Fig S2A–F). Thus, the γδ17 T-cell bias in the pLNs is a universal phenotype upon ageing irrespective of local microbiomes.

To determine whether the biased γδ17 phenotype we observed in aged mice is specific to the pLNs or also common in other secondary lymphoid organs, we investigated the γδ T-cell pool in the mesenteric LN (mLN; Appendix Fig S3A–D) and the spleen (Appendix Fig S4A–E). In both organs, we found an increase in γδ17 T cells and a decline of the γδ1 lineage upon ageing, albeit to a lesser degree compared with pLNs. The proportion of Vγ6+ T cells was also significantly increased, and Vγ1+ T cells were decreased in mLN and spleen from old mice with the changes again being less severe

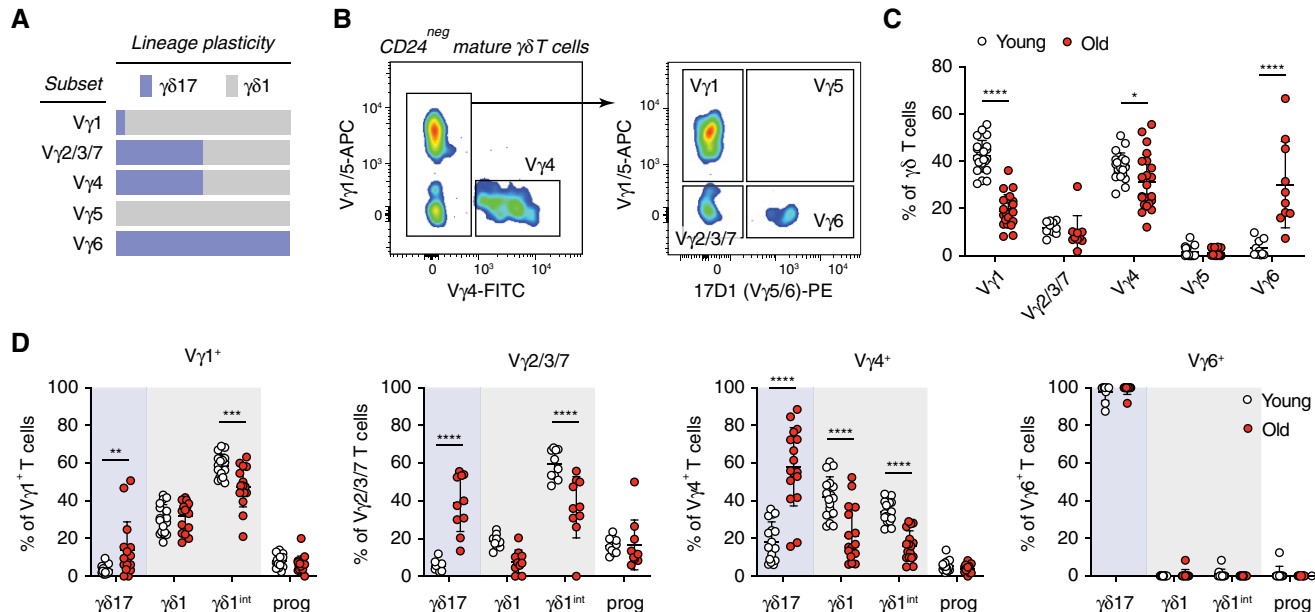

**Figure 2. γδ17-committed Vγ4+ and Vγ6+ cells are the main subsets in pLNs of old mice.**

A  Distinct lineage plasticity of different γδ T-cell subsets according to their TCRγ chain usage.

B  Separation of different γδ T-cell subsets according to their TCRγ chain usage by flow cytometric analysis. The expression of CD45RB, CD44 and CD27 by each γδ T-cell subset was analysed (as in Fig 1 and Appendix Fig S1).

C  Proportion of each γδ T-cell subset in total γδ T cells from pLNs of young and old mice. Results shown are from 23 young and 22 old mice (n = 11 experiments).

D  γδ1 and γδ17 lineage commitment of each γδ T-cell subset in pLNs of young and old mice. Results shown are from 10 pairs of young and old mice (n = 6 experiments).

Data information: Statistical significances for changes in cell proportions were assessed by two-way ANOVA (C and D). Error bars represent SD. *P < 0.05; **P < 0.01; ***P < 0.001; ****P < 0.0001.

---

compared with pLNs. However, the ~ 2-fold increase in the proportion of γδ T cells in the T-cell pool that was observed in the pLN was not seen in mLN and spleen.

**Ageing has minimal impact on the transcriptome of γδ T cells**

In order to determine the mechanism underlying the γδ17 bias in old pLNs, we carried out transcriptome analysis to compare purified Vγ6+ γδ17, Vγ4+ γδ17, Vγ4+ γδ1 and Vγ1+ γδ1 T cells from young and old pLNs (sorting strategy provided in Appendix Fig S5A and B). We confirmed the purity of sorted populations by analysis of characteristic transcription factors, surface markers, cytokines, chemokines and receptors as well as effector molecules, reported to delineate respective γδ1 and γδ17 T-cell subsets (Fig 3A and Appendix Fig S6). Overall, when compared with γδ1 T cells (Vγ1+ and Vγ4+), γδ17 T cells (Vγ4+ and Vγ6+) showed higher expression of *Cd44* and lower expression of *Ptprc*, which are both surface markers used for the segregation of γδ1 and γδ17 T cells by FACS sorting [23]. Consistent with previous reports, Vγ4+ and Vγ6+ γδ17 T cells expressed *Ccr2, Ccr6, Il7r* and *Il23r* at a higher level and down-regulated expression of *Cd27* and *Sell* [8,21,24,25,27,28,35]. Master transcription factors were highly expressed in the respective lineage: *Rorc, Sox13, Maf* and *Zbtab16* in γδ17 T cells and *Eomes, Tbx21* and *Id3* in γδ1 T cells [19,27,36–43]. In homeostasis, γδ1 T cells expressed higher levels of *Ifng* and γδ17 T cells sporadically expressed *Il17a*. Interestingly, Vγ4+ and Vγ6+ γδ17 T cells expressed high levels of TCR complex component *Cd3e* [44] and of

*Tcrg-C3* and *Tcrg-C1*, respectively. By contrast, cytotoxic molecules and NK receptors were highly expressed in Vγ1+ and Vγ4+ γδ1 T cells (Fig 3A and Appendix Fig S6).

Principal component analysis (PCA) revealed distinct separation between γδ1 and γδ17 lineages in PC1 but γδ T cells expressing different Vγ chains were not separated in PC2 (Fig 3B). Notably, γδ T cells from young animals showed higher variance in the γδ1 lineage and cells from old mice showed higher variance in the γδ17 lineage along PC2 (Fig 3B). Nevertheless, direct comparison of each γδ T-cell subset from young and old mice identified only a small number of differentially expressed genes in Vγ4+ γδ1 and γδ17 subsets (Fig 3C). No changes between young and old mice were detected in Vγ6+ γδ17 and Vγ1+ γδ1 T cells.

Since no major functional or transcriptomic changes were detected between young and old γδ T-cell subsets, we investigated whether the increase in γδ17 T cells in old mice could be due to (i) a change in TCR repertoire and/or (ii) changes in the microenvironment of the pLN upon ageing.

**Ageing alters Vδ chain usage and clonal substructure but not global TCR diversity**

The αβ TCR repertoire has been shown to decline with age [45]. We asked whether TCR diversity of γδ T cells from pLNs also changes upon ageing in the variant Vγ4+ and Vγ1+ subsets using invariant Vγ6+ as a control. From the RNA-Seq data (paired-end, 125 bp sequencing) of purified γδ T-cell subsets, we reconstructed CDR3

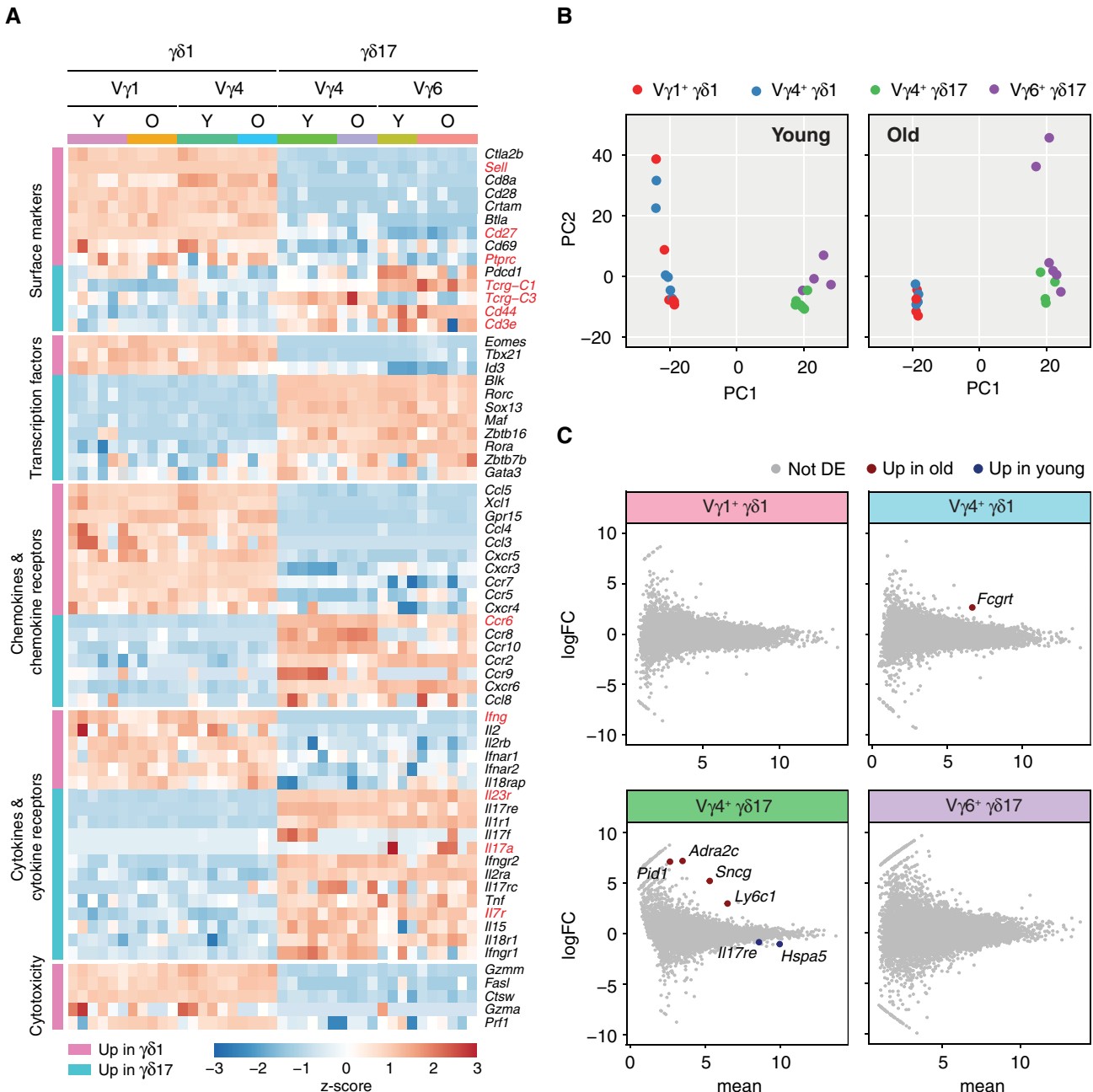

**Figure 3. Transcriptomic analysis identifies minimal differences between γδ T-cell subsets isolated from young and old mice.**

A  Differentially expressed genes between γδ1 and γδ17 lineages were identified by RNA-Seq analysis using *edgeR*. Heat map of relevant genes for γδ1 and γδ17 lineage differentiation and function as well as newly identified genes. Genes shown are grouped by their functions and ranked (top to bottom) by log fold change between γδ1 and γδ17 lineages. The expression of genes marked in red was validated at the protein level by flow cytometry (IFN-γ and IL-17A were examined with or without PMA/ionomycin stimulation *in vitro*; Fig 1 and Appendix Fig S1).

B  Segregation of γδ1 and γδ17 T-cell subsets by principle component analysis (PCA). Each dot represents one RNA-Seq library. Each library is coloured based on the cell subset.

C  Differential expression (DE) analysis using *edgeR* identifies genes upregulated in old (red) and young animals (blue) for each subset. logFC: log fold change in expression.

sequences using MiXCR in the RNA-Seq mode [46,47]. We confirmed the ability of MiXCR to reconstruct the correct Vγ chains for each of the γδ T-cell subsets (Appendix Fig S7A and B). For further downstream analyses, we selected the γδ T-cell subset-specific Vγ

chains. Focusing on variant Vγ1⁺ γδ1, Vγ4⁺ γδ1 and Vγ4⁺ γδ17 cells, we found surprisingly no significant difference in γδ TCR diversity between young and old animals, as indicated by very similar inverse Simpson indexes (Fig 4A).

Next, we assessed the recombination of TCRδ chains and observed distinct preferences in the choice of Vδ segments amongst the different subsets (Fig 4B). In young mice, TCRδ chains utilised by Vγ1$^+$ γδ1 T cells contained mainly rearrangements with Vδ6 (~ 40%) and Vδ2 (~ 30%) segments, followed by Vδ7 (~ 10%), Vδ5 (~ 10%), Vδ4 (~ 7%) and Vδ3 (~ 3%), consistent with a previous study [48]. In old mice, the use of Vδ6 increased slightly to ~ 50% and the use of Vδ2 decreased marginally to ~ 25% compared with young mice (Fig 4B, upper-left panel). In young mice, the majority of TCRδ chains from the Vγ4$^+$ γδ1 T-cell samples contained a Vδ7 segment (~ 75%), followed by Vδ5 (~ 10%), Vδ6 (~ 8%) and Vδ4 (~ 7%) again similar to observations in a previous study [48] (Fig 4B, upper-right panel). In old mice, the preferential use of Vδ7 by Vγ4$^+$ γδ1 T cells was slightly reduced to ~ 70% (Fig 4B, upper-right panel). Strikingly, ageing showed a profound impact on the Vδ segment usage for TCRδ recombination of Vγ4$^+$ γδ17 T cells (Fig 4B, lower-left panel). Vγ4$^+$ γδ17 T cells in young mice preferentially utilised Vδ5 (~ 50%), Vδ4 (~ 30%) and Vδ7 segments (~ 10%), for the recombination of their TCRδ chain. In old mice, the preference amongst Vδ5 and Vδ4 segments was reversed compared with young mice: Vδ4 (~ 60%) was preferentially used followed by Vδ5 (~ 30%; Fig 4B, lower-left panel). As reported, invariant Vγ6$^+$ T cells from young and old mice used only the Vδ1 segment for the assembly of their γδ TCR [44,49,50] (Fig 4B, lower-right panel).

We interrogated the profound changes in TCRδ chain preference observed in Vγ4$^+$ γδ17 T cells by investigating the clonality in Vδ4 and Vδ5 sequences (Fig 4C). Looking at the 10 most frequent Vδ4 clones per individual mouse, we found 6 clones expanded (> 1%) in old mice representing from > 1 to up to 33% of the entire repertoire in the individual mouse. Half of the clonal expansions were private (occurring in 1 out of 4 old mice), while the other half occurred in 2–3 out of the 4 mice. When we looked at the Vδ5 sequences, we also found one expanding clone in 1 out of 4 old mice representing up to 18% of the repertoire. Furthermore, we identified two CDR3 clones from separate mice with different nucleotide sequences both giving rise to an emerging Vδ4$^+$ clone with CALMERDIGGIRATDKLVF amino acid sequence (Fig 4C). Most interestingly, we detected the canonical ASGYIGGIRATDKLV (Vγ4Jγ1/Vδ5Dδ2Jδ1) clone [48] in all individuals and found that this dominant clone in young mice decreases over 50% in 3 out of 4 old mice.

Thus, although organismal ageing did not impact on global γδ TCR diversity, it affected Vδ gene segment usage, led to both private and non-private clonal expansions and a collapse of the recently discovered, dominant invariant ASGYIGGIRATDKLV clone in Vγ4$^+$ γδ17 T cells.

## Increased IL-7 in the LN microenvironment during ageing leads to accumulation of γδ17 T cells

To determine whether the microenvironment affects the γδ17 bias, we interrogated the expression of cytokines associated with activation and homeostatic maintenance of γδ17 T cells. We determined the mRNA expression levels of IL-1β and IL-23, which promote polarisation of γδ17 T cells in peripheral tissues [8,35,51,52], as well as IL-2, IL-15 and IL-7, which are involved in the maintenance of γδ T cells [53–55], in whole pLNs of old and young mice (Fig 5A). The expression of IL-1β, IL-23 and IL-15 was not significantly different between young and old pLNs. IL-2 mRNA expression was low but slightly upregulated in old pLNs. Most strikingly, IL-7 mRNA was highly expressed in the pLNs of both young and old mice and its levels were 3- to 4-fold upregulated in old mice (Fig 5A). Interestingly, IL-7 has been reported to preferentially promote the expansion of IL-17-producing CD27$^{neg}$ γδ T cell in the pLNs upon TCR stimulation [25]. As previously reported [23,26], we found that the expression of IL-7 receptor-α (CD127) is over 2-fold higher in γδ17 compared with γδ1 T cells (Figs 3A and 5B and Appendix Fig S1C).

IL-7 is constitutively secreted by stromal fibroblastic reticular cells in the T-cell zone [56] and by lymphatic endothelial cells [57]. To determine whether IL-7 secreting stroma cells generate a supportive niche for Vγ6$^+$ γδ17 T cells during ageing, we used RNAscope to interrogate the spatial relationship between Vγ6$^+$ γδ17 T cells and IL-7-producing cells. In young and old mice, γδ T cells were mainly localised in the T-cell zone (Fig 5C). Despite clear involution of pLNs in aged mice, the density of γδ T cells, particularly Vγ6$^+$ γδ17 T cells, in T-cell zone per mm$^2$ was highly increased (Fig 5C and D). Consistent with the flow cytometric analysis (Fig 2C), we showed that the proportion of Vγ6$^+$ γδ17 T cells in the γδ T-cell pool is dramatically enriched in aged mice, suggesting the selective accumulation of this unique γδ T-cell subset (Fig 5E).

In pLNs, $Il7$ expression was mainly restricted to T-cell zone where the expression was ~ 6-fold higher compared with the follicle. Importantly, we found that the T-cell zone in the old pLNs contained ~ 5-fold more $Il7$ mRNA compared with young pLNs (Fig 5F). In both old and young mice, γδ T cells localised to the T-cell zone of the pLNs, with only a few cells found in the periphery of follicles (Fig 5G). Strikingly, all γδ T cells were localised in close proximity to IL-7 mRNA expressing cells (on average 20 μm) and this distance was reduced to 10 μm for Vγ6$^+$ T cells in old pLN (Fig 5H).

To determine whether IL-7 is indeed functionally important for the expansion of γδ17 T cells, in particular Vγ6$^+$ γδ17 T cells, in the pLNs, we treated young mice with either isotype control or IL-7-neutralising antibodies and administered EdU for 3 days to assess proliferation (Fig 5I). Consistent with a previous study [58], γδ17 T cells incorporated more EdU than γδ1 T cells in the pLNs (Fig 5J). In particular, we found that Vγ6$^+$ T cells were most proliferative amongst all γδ T-cell subsets with 50% of cells having incorporated EdU while < 10% of Vγ1$^+$, Vγ2/3/7 and Vγ4$^+$ cells labelled positive for EdU (Fig 5K). Most strikingly, $in vivo$ proliferation of γδ17 T cells but not γδ1 T cells was diminished selectively by IL-7 neutralisation (Fig 5J) and Vγ6$^+$ T cells were the main subset relying on IL-7 signalling for proliferation (Fig 5K). Of note, the proliferation of γδ17 T cells within Vγ4$^+$ T cells was also affected by IL-7 neutralisation, but to a lesser degree (Fig EV2A). CD4$^+$ and CD8$^+$ T cells in pLNs of young mice were not affected by short-term $in vivo$ IL-7 neutralisation (Fig EV2B and C).

We performed the same experiment in mid-aged mice (12 months old). Although the overall incorporation of EdU in all T cells was significantly lower in older mice, we repeated our observations from young mice (Appendix Fig S8A–F). The proliferation of γδ17 T cells but not γδ1 T cells was dependent on IL-7 (Appendix Fig S8A). Vγ6$^+$ T cells remained the most proliferative γδ T-cell subset and were the only subset dependent on IL-7 for proliferation (Appendix Fig S8B), suggesting that Vγ6$^+$ T cells could

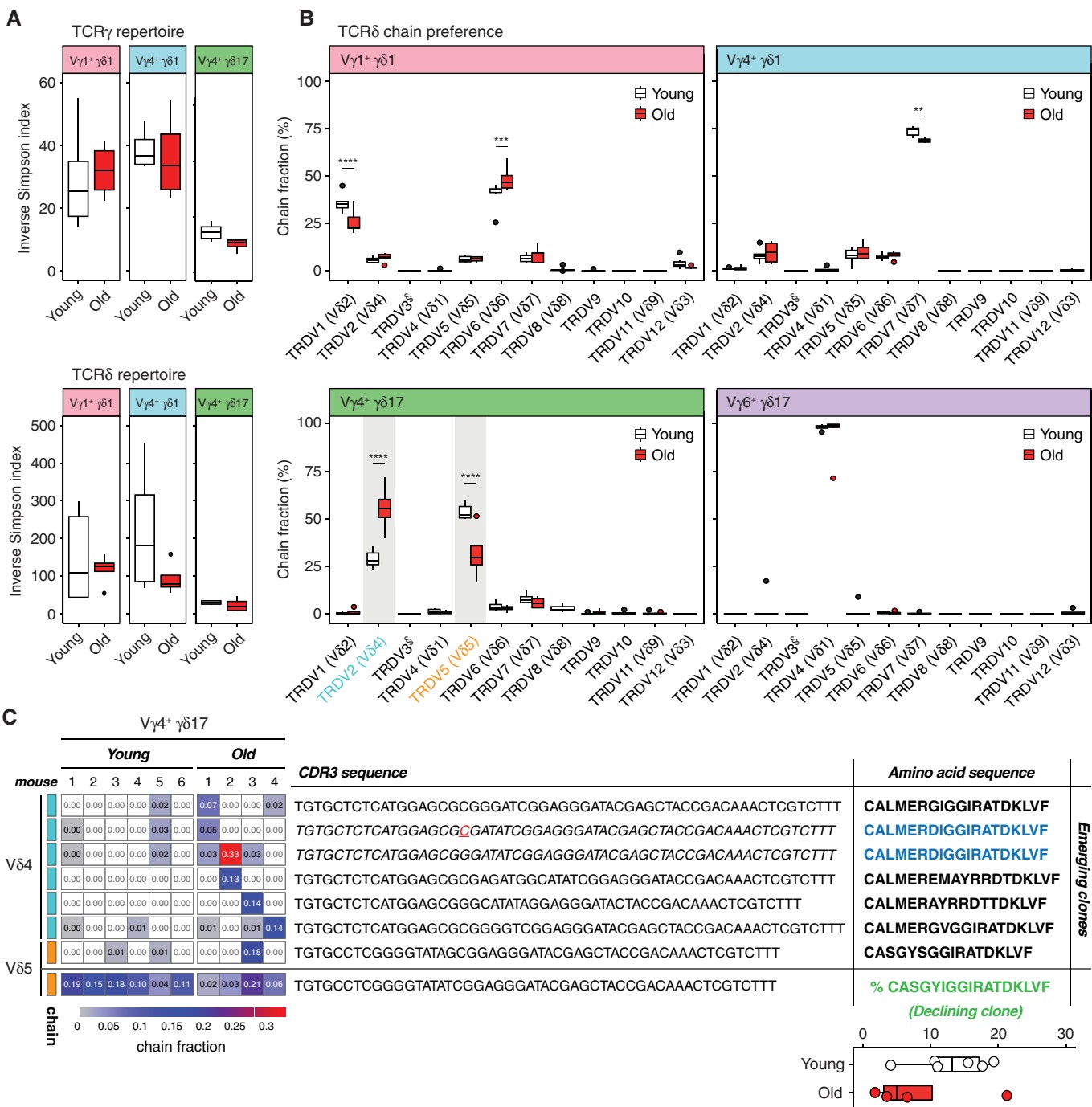

**Figure 4. γδ TCR repertoire analysis and Vδ segment usage of γδ T-cell subsets in pLNs of young and old mice.**

A  The diversity of TCRγ (top) and TCRδ (bottom) repertoires was evaluated in sorted Vγ1⁺ γδ1, Vγ4⁺ γδ1 and Vγ4⁺ γδ17 subsets of young and old mice and is represented as inverse Simpsons index.

B  Vδ chain usage in Vγ1⁺ γδ1, Vγ4⁺ γδ1 and Vγ4⁺ γδ17 and Vγ6⁺ γδ17 T-cell subsets from young and old mice (§TRDV3 is a pseudogene). The fraction of in-frame rearrangements of Vδ gene segments within the sorted populations is shown.

C  Emerging and declining clones defined by CDR3 nucleotide sequence in old mice. Heat map and indicated frequency show the abundance of specific clones in each young and old individual. (Inset right) Percentage of the canonical CDR3 amino acid sequence CASGYIGGIRATDKLVF in sorted Vγ4⁺ γδ17 T cells from the pLNs of young and old mice. IMGT gene names and their corresponding TCR Vδ chains are summarised in Appendix Table S2.

Data information: Data shown are from 4 to 6 mice/condition (three independent experiments). Statistical significances for changes were assessed by Mann–Whitney test (A and C) or two-way ANOVA (B). In the box plots (A–C), lower and upper hinges indicate the first and third quartile, and the horizontal line within the box indicates the median. Upper whiskers extend from Q3 to 1.5× IQR (inter quartile range) and lower whiskers from Q1 to 1.5× IQR. Outliers identified by Tukey's rule were plotted individually (A and B). **$P < 0.01$; ***$P < 0.001$; ****$P < 0.0001$.

gradually outgrow other γδ T cells in the pLNs. Due to the short time period of IL-7 blockade and the overall reduced proliferation in the pLNs of mid-age mice, the proportion of Vγ6⁺ T cells was only slightly reduced as a result of IL-7 neutralisation (Appendix Fig S8C). Interestingly, γδ17 T cells within the Vγ1⁺ and Vγ4⁺ subsets were also affected by IL-7 neutralisation (Appendix Fig S8D). The proliferation of CD4⁺ and CD8⁺ T cells in pLNs remained unaffected by IL-7 blockade (Appendix Fig S8E and F). These results

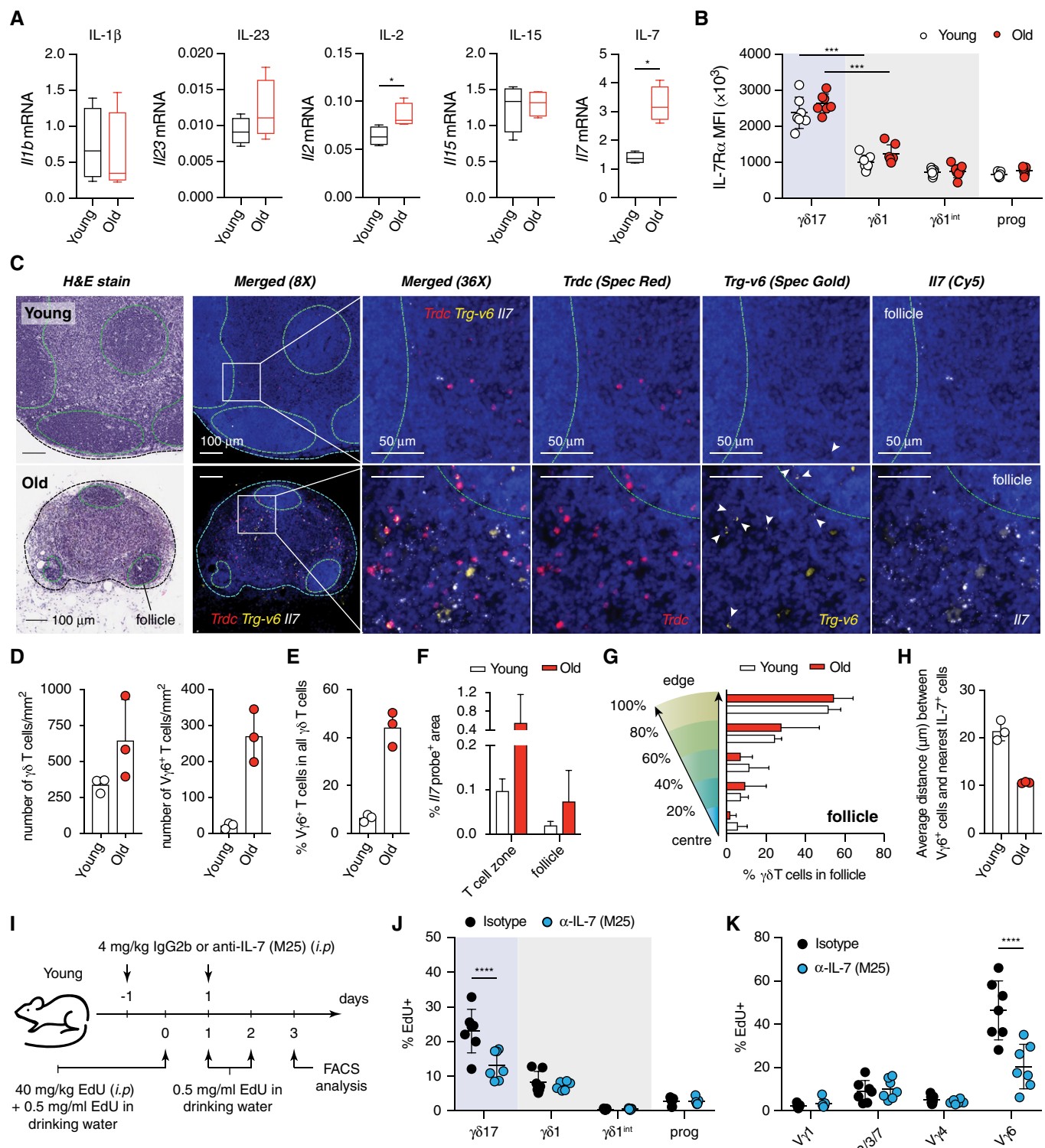

**Figure 5.**

**Figure 5.  IL-7 is highly expressed in pLNs of old mice and creates a niche for Vγ6⁺ γδ17 T-cell expansion.**

A      The expression of IL-1β, IL-23, IL-2, IL-15 and IL-7 mRNA in the pLNs of young and old mice was analysed by qRT–PCR and normalised to *Tbp* as a housekeeping
       gene. Similar results were obtained using *Hprt* and *B2m* as housekeeping genes. Results are representative of two independent experiments each with four young
       and four old mice.
B      Protein expression of IL-7Rα by γδ1 and γδ17 T cells from young and old mice. Results shown are from seven young and seven old mice (*n* = 3 experiments).
C      Serial sections of inguinal LNs from young (top) and old (bottom) were stained with H&E or specific probes targeting the constant region of TCRδ, Vγ6 TCR and
       IL-7 mRNA as indicated. Representative images shown are from three pairs of young and old mice. Arrowheads indicate cells stained positive for Vγ6 TCR.
D–H   (D) Density of total γδ T cells and Vγ6⁺ T cells, (E) proportion of Vγ6⁺ T cells of total γδ T cells, (F) expression of IL-7 mRNA in the T-cell zone and in the follicle, (G)
       localisation of γδ T cells in the follicles and (H) average distance between Vγ6⁺ T cells and the nearest of IL-7-producing cells were quantified by analysis of images
       using Halo software (Indica Lab). Results are representative of 2 independent experiments each with 3 pairs of young and old mice.
I      Experimental design of IL-7 neutralisation *in vivo* with EdU pulsing.
J, K   *In vivo* proliferation of γδ T cells from different lineages (J) and different γδ T-cell subsets (K) in the pLNs of young mice treated with either isotype IgG2b or anti-
       IL-7 neutralising antibody. Results are from two independent experiments with 14 young mice (seven each for control and experimental groups).

Data information: Statistical significances for changes in expression levels were assessed by Mann–Whitney test (A, D, E and H) or two-way ANOVA (B, J and K). Error bars
represent SD.  In the box plots (A), lower and upper hinges indicate the first and third quartile, and the horizontal line within the box indicates the median. Upper
whiskers extend from Q3 to the maximum and lower whiskers from Q1 to the minimum value. *$P < 0.05$; ***$P < 0.001$; ****$P < 0.0001$.

confirm that the increase in Vγ6⁺ γδ17 T cells in the pLNs upon ageing is regulated by increased local IL-7.

Taken together, we show that IL-7 production in the T-cell zone of pLNs is highly increased upon ageing and leads to a skewed peripheral γδ T-cell pool with the enrichment of γδ17 T cells, especially Vγ6⁺ T cells, which might favour pro-inflammatory immune responses.

### γδ17 T-cell bias augments tumour growth in aged mice

γδ T cells have important and well-established anti-tumour roles due to cytotoxic function and IFN-γ secretion of γδ1 T cells. By contrast, γδ17 T cells have been shown to mediate pro-tumour activities [10,11]. We hypothesised that LN-resident γδ T cells can be activated upon tumour challenge, migrate into the tumour mass and impact on the tumour microenvironment (TME).

First, we tested whether LN-resident γδ T cells can infiltrate into tumours using the 3LL-A9 syngeneic Lewis lung cancer model. We blocked T-cell egress from LNs by administering FTY720 to mice (Fig 6A) [20,22]. As expected, FTY720 treatment reduced the number of CD4⁺ and CD8⁺ T cells in the tumour (Fig EV3A and B). Strikingly, also the number of γδ T cells was greatly reduced to below 20% upon FTY720 treatment compared with control animals (Fig 6A). In addition, the composition of the γδ T-cell pool was altered by FTY720 treatment. Progenitor γδ T cells increased, while γδ1 T cells decreased. On the γδ T-cell subset level, Vγ1⁺ and Vγ6⁺ T cells declined, and Vγ2/3/7 and Vγ4⁺ γδ T cells increased, suggesting different levels of egress from the pLNs (Fig EV3C and D). Taken together, we show for the first time that LN-resident γδ T cells can contribute significantly to the γδ T-cell pool in tumours.

Next, we asked whether the γδ17-biased γδ T-cell pool in old mice can affect the tumour response. Strikingly, we found that 3LL-A9 tumours grew faster in old mice (Fig 6B and C). γδ T-cell infiltration into the tumour was similar in young and old mice (Fig EV4A and B), but the balance between γδ1 and γδ17 T cells was altered: while tumours from young mice maintained a substantial proportion of anti-tumour γδ1 T cells, over 90% of the γδ T-cell pool in tumours of aged mice were tumour-promoting γδ17 T cells (Fig 6D). The tumour-infiltrating γδ T-cell pool in young mice was heterogeneous, containing Vγ1⁺, Vγ2/3/7, Vγ4⁺ and Vγ6⁺ T cells. In contrast, the tumour-infiltrating γδ T-cell pool in old mice consisted mainly of Vγ6⁺ T cells (> 80%; Fig 6E). Importantly, the proportion

of Vγ6⁺ T cells in total tumour-infiltrating γδ T-cell pool correlated positively with tumour size (Fig 6F). Skin-resident Vγ5⁺ T cells [59] were absent from the subcutaneous tumours of old and young mice. In the tumour, lineage bias of subsets was very different from the homeostasis observed in the pLNs. In the tumour microenvironment, progenitor and γδ1ⁱⁿᵗ populations were lost, Vγ1⁺ and surprisingly Vγ4⁺ T cells were γδ1 and only Vγ2/3/7 and Vγ6⁺ T cells were γδ17-committed (Fig EV4C).

We then asked which cells were activated and/or exhausted (by their PD-1 and Tim-3 expression) in the TME to determine the involvement of different subsets in the anti-tumour response (Figs 6G and H, and EV4D). Approximately 50% of γδ T cells in tumours of young mice and 70% of γδ T cells in tumours of old mice were highly activated and exhausted (PD-1⁺, Tim-3⁺; Fig 6G). Interestingly, only γδ17 T cells showed high levels of activation (Fig EV4D), while γδ1-committed Vγ1⁺ and Vγ4⁺ T cells were not activated (Fig 6H). The majority of tumour-infiltrating Vγ2/3/7 T cells in both young and old mice were single-positive for Tim-3⁺. Most intriguingly, only tumour-infiltrating Vγ6⁺ T cells were highly activated/exhausted with high expression levels of PD-1 and Tim-3 (Fig 6H).

Neutrophils can limit tumour growth by inhibiting the pro-tumour function of γδ17 T cells [60,61]. In order to see whether different amounts of neutrophil infiltration could account for the different tumour growth observed in young and old mice, we assessed the presence of neutrophils in the TME. No difference in the infiltration of Ly6Gʰⁱ CD11b⁺ Ly6Cⁱⁿᵗ neutrophils was observed between tumours of young and old mice (Fig 6I).

Next, we confirmed that pro-tumour IL-17 is indeed produced by γδ17 T cells in the TME. We found that 20–40% of γδ T cells from the tumour produced IL-17 upon *ex vivo* restimulation, representing 60–80% of all IL-17-producing cells in the TME (Appendix Fig S9A and B). Amongst the γδ T-cell subsets, both Vγ4⁺ and Vγ6⁺ T cells produced IL-17, with the Vγ6⁺ subset containing the highest proportion of IL-17 producers (Appendix Fig S9C). The level of IL-17 production by Vγ6⁺ T cells was reduced in the TME of old mice, likely due to the more exhausted status of the cells at the point of analysis (Fig 6H).

In the tumour-draining LN, the overall lineage commitment and subset composition of γδ T cells were similar to the steady state in young and old mice (Fig EV5A and B). The γδ17 bias within Vγ2/3/7 and Vγ4⁺ T-cell subsets during ageing was not observed in the

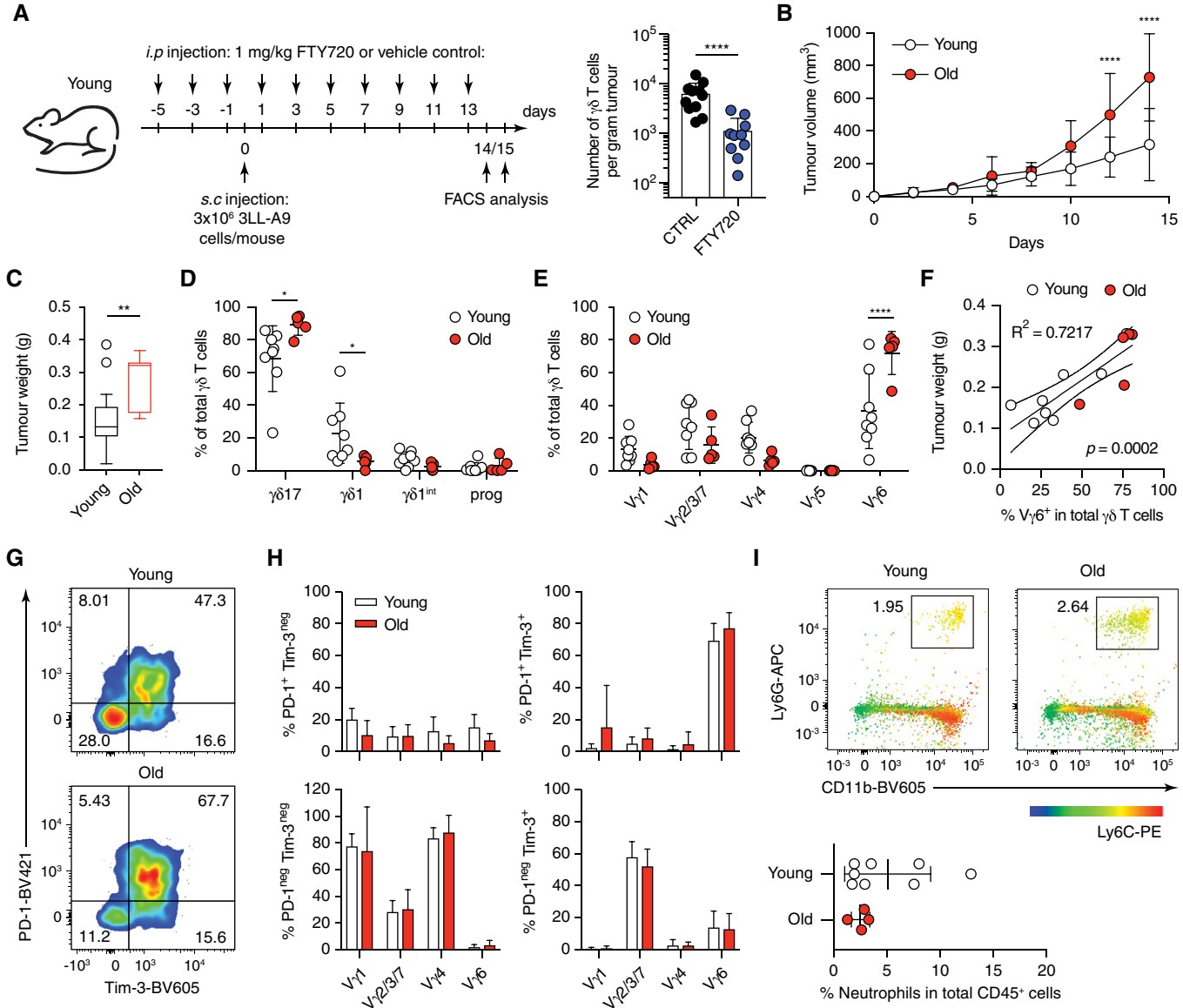

**Figure 6.  Selectively activated Vγ6+ γδ17 T cells infiltrated from draining lymph nodes into tumour correlates with faster tumour growth in old mice.**

A    Young mice were injected every other day by *i.p.* with FTY720 at a dose of 1 mg/kg or with vehicle control containing 2.5% ethanol and 2% β-cyclodextrin from day −5 to day 13. 3 × 10^6 3LL-A9 cells were given to control and FTY720-treated mice by subcutaneous injection on day 0. Tumours were harvested at day 14 or 15 for FACS analysis. The infiltration of γδ T cells from draining LN into tumour was blocked in young mice by the FTY720 treatment.

B    3LL-A9 Lewis lung carcinoma cells were injected subcutaneously into young and old C57BL/6 mice, and tumours were analysed 14 days postinjection. Tumour growth curves shown are from 21 young mice (*n* = 4 experiments) and seven old mice (*n* = 3 experiments).

C    Tumour weights shown are from 23 young (*n* = 5 experiments) and seven old mice (*n* = 3 experiments).

D    γδ1 and γδ17 lineage commitment of total γδ T cells within the tumour of young and old mice. Results are from eight young and five old mice (*n* = 2 experiments).

E    γδ T-cell subsets recovered from the tumour. Results are from eight young and five old mice (*n* = 2 experiments).

F    Linear regression fit between the weight of tumours and the proportion of Vγ6+ T cells in total tumour-infiltrating γδ T cells. Results are from eight young and five old mice (*n* = 2 experiments).

G, H   Activation and exhaustion of total tumour-infiltrating γδ T cells (G) and different γδ T-cell subsets (H) in young and old mice. FACS files acquired for each individual mouse were concatenated for analysis, and the results were shown as representative dot plots. Results are from eight young and five old mice (*n* = 2 experiments). Cell populations with the total cell number < 10 were excluded from the analysis.

I    Proportion of neutrophils (Ly6G+ CD11b+ Ly6C^int) in total CD45+ tumour-infiltrating immune cells of young and old mice. Results are from eight young and four old mice (*n* = 2 experiments).

Data information: Statistical significances for differences were assessed by Mann–Whitney test (A, C and I) or two-way ANOVA (B, D, E and H). Error bars represent SD.  In the box plots (C), lower and upper hinges indicate the first and third quartile, and the horizontal line within the box indicates the median. Upper whiskers extend from Q3 to 1.5× IQR (inter quartile range) and lower whiskers from Q1 to 1.5× IQR. Outliers identified by Tukey's rule were plotted individually. *P < 0.05; **P < 0.01; ****P < 0.0001.

tumour-draining LN of old mice (Fig EV5C). At the point of analysis, no activation of Vγ1$^+$, Vγ2/3/7 and Vγ4$^+$ T cells was detected (Fig EV5D). Importantly, only Vγ6$^+$ T cells expressed Tim-3 and PD-1 in old and young mice (Fig EV5D). No expression of Tim-3 or PD-1 by Vγ6$^+$ T cells was observed in the pLNs under homeostatic conditions (Appendix Fig S10). These results suggest that Vγ6$^+$ T cells become activated in the tumour-draining LN. Interestingly, the low number of Tim-3 single-positive Vγ2/3/7 γδ17 T cells observed in the tumour cannot be found in the dLN, indicating a LN-independent origin of activation for this subset. These results were confirmed by the use of another syngeneic Lewis lung tumour model (Appendix Fig S11A–E).

Taken together, we show that Vγ6$^+$ γδ T cells become selectively activated in the draining LN, migrate into the tumour and represent the majority of the tumour-resident γδ T cells in old mice. The biased γδ17 T-cell pool in pLNs under homeostatic condition upon ageing augments the infiltration of pro-tumour Vγ6$^+$ γδ T cells, which is associated with enhanced tumour growth in aged mice.

## Discussion

How γδ T cells change upon organismal ageing has not been extensively explored. We have conducted the first comprehensive study of the murine γδ T-cell compartment in pLNs during ageing. Remarkably, we found that—upon ageing—the γδ17 lineage dominates the γδ T-cell pool at the expense of γδ1 T cells. The striking γδ17 bias with age is due, predominantly, to the accumulation of Vγ6$^+$ T cells, and in part by increased γδ17 bias of Vγ4$^+$ and Vγ2/3/7 T-cell subsets.

The transcriptome of γδ T cells from young and old mice showed only minimal differences in gene expression. We found that Ly6C is upregulated in Vγ4$^+$ γδ17 T cells from aged mice. Previous work has determined that cross-linking of Ly6C induces LFA clustering/adhesion, thereby supporting the homing of naïve and central memory CD8$^+$ T cells to the lymph node [62,63]. Whether this is a contributing factor enabling Vγ4$^+$ γδ17 T cells to accumulate in the pLNs requires further investigation. In addition, old γδ T cells secrete the same level of IFN-γ and IL-17 as their young counterparts. Similar results have been observed for naïve CD4$^+$ T cells at the transcriptional and functional level [64]. Our results thus support the notion that γδ T cells are intrinsically unaffected by ageing, and instead, the age-related γδ17 bias is a function of the altered environment, for instance, as we have shown, in the pLNs.

For the first time, we localised γδ T cells, especially Vγ6$^+$ T cells, in pLNs of old mice and found that the majority of murine γδ T cells reside in the T-cell zone. γδ T cells were also sporadically observed in subcapsular and medullary sinuses as previously described [65,66] but at a low level.

While peripheral γδ1 T cells are mainly replenished by thymic output, γδ17 T cells are thought to be derived exclusively from foetal thymus and maintained by proliferation and self-renewal in the periphery [58,67]. Only under certain circumstances—involving TCR stimulation in the presence of IL-1β and IL-23—can γδ17 T cells develop in adult mice [35,51,52]. We did not find increased levels of IL-1β or IL-23 in the pLN of old mice but discovered that IL-7 is highly expressed in the aged pLNs and the number of γδ T cells correlates with the amount of IL-7. γδ T cells are found in close

proximity to IL-7 (on average < 20 μm in young and < 10 μm in old mice), indicating that IL-7-producing cells are creating a niche in which IL-7Rα$^{hi}$ Vγ6$^+$ γδ17 T cells can be maintained in old mice. By neutralising IL-7 *in vivo*, we were able to functionally proof that γδ17 T cells and in particular Vγ6$^+$ T cells rely on IL-7 for proliferation in the pLNs. Further investigations should determine whether a reduced thymic output in old mice is partially responsible for the decrease in γδ1 T cells in the pLN, and to what extent an increased IL-7 production by aged stroma cells and a reduction in IL-7-consuming immune cells contribute to increased IL-7 levels in the aged pLN favouring expansion of Vγ6$^+$ T cells.

We next characterised the γδ TCR repertoire of variant subsets in young and old mice in order to elucidate changes that lead to a reduced γδ1 T-cell pool and expansions in γδ17 cells. In contrast to previous work demonstrating the collapse of TCR diversity in αβ T cells upon ageing [45], the TCR diversity does not collapse in γδ T cells of aged mice. TCR diversity was higher in the γδ1 subset compared with γδ17. Interestingly, Vδ chain usage was altered especially in variable Vγ4$^+$ γδ17 T cells upon ageing. Analysis of the CDR3 regions of the altered Vδ chains revealed private and semi-private clonal expansions within the Vδ4 and Vδ5 repertoire. Interestingly, we observed that the innate Vγ4Jγ1/Vδ5Dδ2Jδ1 (ASGYIGGIRATDKLV) clone in the Vδ5 repertoire [48] declined in 3 out of 4 old animals analysed. Taken together, we have shown that consequences of ageing on the γδ TCR repertoire are as follows: (i) altered TCRδ chain usage; (ii) clonal expansions of γδ17 clones perhaps indicating the appearance of age-related antigens; and (iii) the loss of a recently described invariant innate clone, which indicates the loss of a specific γδ T-cell reactivity upon ageing.

We show that γδ17 T cells in the pLNs are highly proliferative compared with other γδ and αβ T-cell subsets under homeostatic conditions, suggesting that the LN-resident and/or LN-recirculating γδ17 T cells could represent a specialised population with unique proliferative and activation features. In the context of cancer, the role of the LN-resident γδ T-cell pool has not been explored. In tumours, γδ T cells are the main source of IFN-γ at the early stage of tumour development in young mice [6]. We asked whether the acquired γδ17 bias in pLNs during ageing would impact on the early tumour microenvironment. Using a Lewis lung carcinoma model and blocking egress of T cells from LNs using FTY720, we found that γδ T cells egressing from pLNs are constituting the majority of the γδ T-cell pool in the tumour. Importantly, Vγ6$^+$ T cells but not any other γδ T-cell subset become activated in the tumour-draining LN. Due to the high number of these pro-tumourigenic cells in the pLNs, the tumour microenvironment becomes highly tumour-promoting and tumours progress faster in old mice. Interestingly, γδ17-committed Vγ4$^+$ and Vγ2/3/7 T cells are not activated upon tumour challenge, indicating that only the invariant Vγ6 TCR can recognise tumour-associated antigens or other signals, at least in the 3LL-A9 model.

Ageing is associated with chronic inflammation resulting from systemically increased pro-inflammatory cytokines. This predisposition to inflammatory responses can significantly affect the outcome of infection [68] and cancer immunotherapy [69]. In aged mice, an increase in Th17-polarised CD4$^+$ T cells [70,71], as well as higher IL-17 secretion by liver-resident NKT cells, has been described [68]. Here, we discovered γδ17 T cells as a new critical pathogenic player during ageing.

Interestingly, a shift towards more effector phenotypes, largely maintained TCR diversity and a change in Vγ/Vδ usage has also been detected in the peripheral blood of humans upon organismal ageing [72,73], suggesting similar age-related processes occurring in the murine and human γδ T-cell pool.

Taken together, we have identified a novel age-dependent dysregulation of the γδ T-cell pool that is associated with enhanced tumour progression in old mice. Development of therapeutics specifically targeting γδ17 T cells and correcting the biased γδ T-cell pool in the elderly might reduce the susceptibility to age-related diseases including infection and cancer.

# Materials and Methods

### Mice

C57BL/6 mice were purchased from Charles River UK Ltd (Margate, UK) and housed under specific pathogen-free conditions at the University of Cambridge, CRUK Cambridge Institute in accordance with UK Home Office regulations. Mice from a second ageing cohort were bred and maintained in the Babraham Institute Biological Support Unit. Young and old mice from the Babraham cohort received three immunisations by oral gavage 6 days apart with 200 μl PBS containing 37.5 μg/ml CTx (CTx; Sigma-Aldrich #C8052) plus 37.5 μg/ml NP-CTx. Mice were harvested 7 days after the last oral gavage with CTx/NP-CTx. All animals were euthanised in accordance with Schedule 1 of the Animals (Scientific Procedures) Act 1986. Every mouse used was macroscopically examined externally and internally, and animals with lesions or phenotypic alterations were excluded from the analysis.

### Tissue processing and flow cytometry

Peripheral lymph nodes (inguinal and axillary, alone or pooled) and spleen were collected from young and old mice, respectively, mashed through a 40-μm (thymus and pLNs) or a 70-μm cell strainer (spleen; Greiner bio-one) with the plunger of a 2-ml syringe to prepare single-cell suspensions. Cells were washed with PBS once and stained with Fixable Viability Dye eFluor™ 780 (Thermo Fisher Scientific). Fc receptors were blocked with TruStain fcX™ (anti-mouse FCGR3/CD16-FCGR2B/ CD32, clone 93; BioLegend) in FACS buffer containing 3% FCS (Biosera) and 0.05% sodium azide (Sigma-Aldrich) in Dulbecco's phosphate-buffered saline (DPBS; Gibco). Subsequently, cells were stained in FACS buffer with fluorochrome-conjugated antibodies against cell surface antigens (Appendix Table S1).

For the characterisation of Vγ6$^+$ T cells, the staining procedure was modified as follows. Before staining of cell surface markers, cells were stained with GL3 antibodies against TCR Vδ followed by 17D1 hybridoma supernatant (kindly provided by Prof. Adrian Hayday, The Francis Crick Institute, London) that recognises both Vγ5 and Vγ6 TCR. PE-conjugated mouse anti-rat IgM monoclonal antibody (RM-7B4, eBioscience) was then used to capture cells stained positive with 17D1 hybridoma supernatant. Cells were analysed using a FACS LSR II, FORTESSA or ARIA (BD) instrument and FlowJo software (v10.4, FlowJo, LLC).

### *In vitro* stimulation

Single-cell suspensions from peripheral LNs were washed twice with complete RPMI medium [RPMI-1640 (Gibco), supplemented with 10% heat-inactivated FCS (Biosera), 1 mM sodium pyruvate (Gibco), 10 mM HEPES (Sigma), 100 U/ml penicillin/streptomycin (Gibco) and 50 μM β-mercaptoethanol (Gibco)] and plated in 96-well plate with or without 50 ng/ml PMA (Sigma-Aldrich) and 1 μg/ml ionomycin (Sigma-Aldrich) in the presence of GolgiStop (1:1,500 dilution, BD) for 4 h. After incubation, cells were washed once with PBS and stained with Fixable Viability Dye eFluor™ 780 (Thermo Fisher Scientific) followed by blocking with TruStain fcX™ (anti-mouse FCGR3/CD16-FCGR2B/CD32, clone 93; BioLegend) and staining with fluorophore-conjugated antibodies against cell surface antigens in FACS buffer. Cells were then fixed and permeabilised using BD Cytofix/Cytoperm™ Plus Kit for intra-cellular staining with fluorochrome-conjugated antibodies against IFN-γ (clone XMG1.2, BioLegend) and IL-17A (clone TC11.18H10.1, BioLegend). Stained cells were run on a BD FACS LSR II cytometer, and analysis was performed using FlowJo software (v10.4, FlowJo, LLC).

### Isolation of γδ1 and γδ17 T cells

Single-cell suspensions were prepared from inguinal and axillary LNs collected of young and old mice. To enrich γδ T cells, αβ T cells and B cells were depleted from single-cell suspensions by MACS using a biotinylated antibody against TCRβ with anti-biotin microbeads and anti-CD19 microbeads, respectively. Enriched γδ T cells were then stained for FACS sorting as described above. Gating strategy used to identify Vγ1$^+$ γδ1, Vγ4$^+$ γδ1, Vγ4$^+$ γδ17 and Vγ6$^+$ γδ17 T cells is summarised in Appendix Fig S5A and B. γδ T-cell subsets were sorted with a BD FACS ARIA instrument directly into 3 μl of lysis buffer from SMART-Seq v4 Ultra Low Input RNA Kit (1 μl of 10× Reaction Buffer and 2 μl of water) accordingly to the instructions of the manufacturer (Clontech). Cells were centrifuged, immediately frozen in liquid nitrogen and stored at −80°C.

### RNA-Seq library preparation and sequencing

RNA-Seq libraries were prepared using the SMART-Seq v4 Ultra Low Input RNA Kit (Clontech). Cells frozen in lysis buffers were directly complemented with cold nuclease-free water plus RNAse inhibitor (2 U/μl; Clontech) up to 9.5 μl of total volume. The volume of water was estimated calculating the number of events sorted in the BD FACS ARIA and the average drop size for the 70-μm nozzle used (~ 1 nl/droplet). ERCC spike-in RNA (Ambion; 1 μl diluted at 1:300,000) and 3′ SMART-Seq CDS Primer II A (12 μM) were added to the lysis mix. cDNA was prepared following the SMART-Seq v4 Ultra Low Input RNA Kit protocol (Clontech).

After cDNA preparation, RNA-Seq libraries were prepared using Illumina Nextera XT Sample Preparation Kit (Illumina, Inc., USA) and the 96 Index Kit (Illumina, Inc., USA). As previously described, libraries were prepared by scaling down the reactions one-fourth of the manufacturer instructions [74] and libraries were sequenced using paired-end 125 bp sequencing on Illumina HiSeq4000.

## Read alignment of RNA-Seq data

Prior to read alignment, the *Mus musculus* genome (GRCm38) was concatenated with the sequence of ERCC spike-ins (available at http://tools.lifetechnologies.com/content/sfs/manuals/ERCC92.zip). Sequenced reads were aligned against this reference using *gsnap* version 2015-12-31 [75] with default settings. Gene-level transcript counts were obtained using HTSeq version 0.6.1p1 [76] with the -s option set to "no" and using the GRCm38.88 genomic annotation file concatenated with the ERCC annotation file.

## Quality control of RNA-Seq libraries

We excluded libraries with fewer than 40% of reads mapped to annotated exons or fewer than 100,000 total reads. Furthermore, we removed libraries with fewer than 10,000 genes detected with at least one count.

## Normalisation of RNA-Seq libraries

We used the Bioconductor R package *edgeR* [77] for data normalisation. More specifically, we used the *calcNormFactors* to estimate normalisation factors and computed counts per million using the *cpm* function implemented in *edgeR*. Gene-level transcript counts are visualised as *Z*-score scaled, normalised counts.

## Alignment of reads to T-cell receptor genes

TCR repertoire analysis was performed using the MiXCR software [46,47]. In the first step, sequencing reads were aligned to the V, D, J and C genes of the T-cell receptor. For this, we used the *align* function with following settings:
-p rna-seq -s mmu -OallowPartialAlignments = true.

## TCR assembly

The T-cell receptor sequences were assembled by calling the *assemblePartial* function twice to assemble partially aligned sequences. To extend TCR alignments, the *extendAlignments* function was called. In the last step, the *assemble* function was used to fully assemble the V, D, J and C genes of the TCR.

## Exporting individual clones after TCR assembly

Individual clones were collected using the *exportClones* function while excluding out-of-frame variants (-o option) and stop codon-containing variants (-t option). Clones were collected for the different chains (TRD, TRG, TRA, TRB, IGH and IGL) separately. This function returns the count, fraction and information on the V, D, J and C chain of the individual clones per library as defined through their CDR3 nucleotide sequence.

## Quantitative RT–PCR

Thymus, spleen and peripheral LNs (both inguinal and axillary) were collected from healthy young and old mice and homogenised with Precellys 1.4-mm ceramic beads in 2-ml tubes (KT03961-1-003.2, Bertin Instruments) using Precellys 24 lysis and homogenisation unit (Bertin Instruments). Total RNA was extracted from homogenised samples using Ambion PureLink RNA Kit (12183025, Invitrogen) according to the manufacturer's instructions. RNA was quantitated using the NanoDrop Spectrophotometer ND-1000 and diluted in RNase-free water to 100 ng/μl for analysis. Quantitative RT-PCR was carried out using the Superscript III Platinum One-Step qRT-PCR Kit (Life Technologies) and TaqMan™ Gene Expression Assays (Fam) (Life Technologies) to quantify the expression of following genes: *Tbp* (assay ID: Mm00446971_m1), *Hprt* (Mm03024075), *B2m* (Mm00437762_m1), *Btn1a1* (Mm00516333_m1), *Btnl1* (Mm01281669_m1), *Btnl2* (Mm01281666_m1), *Btnl4* (Mm03413106_g1), *Btnl6* (Mm01617956_mH), *Btnl9* (Mm00555612_m1), *Skint1* (Mm01720691_m1), *Il1b* (Mm00434228_m1), *Il2* (Mm00434256_m1), *Il7* (Mm01295803_m1), *Il15* (Mm00434210_m1), *Il17a* (Mm00439618_m1 IL17a) and *Il23a* (Mm00518984_m1). qRT-PCR was performed using a QuantStudio 6 Flex Real-Time PCR System (Thermo Fisher Scientific). For reverse transcription, the thermal cycler was set at 50°C for 15 min followed by a 2-min incubation at 95°C, after which 50 PCR cycles of 15 s at 95°C followed by 1 min at 60°C were run. All samples were run in triplicates, and similar results were obtained for all housekeeping genes used (*Tbp*, *Hprt* and *B2m*).

## RNAscope

Inguinal LNs were isolated from young and old mice, respectively, fixed in 10% NBF (Pioneer Research chemicals Ltd) for 24 h, transferred into 70% ethanol for 24 h and embedded into paraffin blocks. Paraffin sections were cut at 3 mm onto Superfrost plus slides and baked for 1 h at 60°C. Probes and Kits (RNAscope LS Multiplex Reagent Kit, Cat# 322800 and RNAscope LS 4-Plex Ancillary Kit Multiplex Reagent Kit Cat# 322830) were obtained from Advanced Cell Diagnostics. TSA Plus Fluorescein System for 50–150 Slides (Cat# NEL741001KT), TSA Plus Cyanine 3 System for 50–150 Slides (Cat# NEL744001KT), TSA Plus Cyanine 5 System for 50–150 Slides (Cat# NEL745001KT) and Opal 620 Reagent Pack (Cat# FP1495001KT) were from Perkin Elmer. Probes (automated Assay for Leica Systems) and reference sequences were as follows: RNAscope LS 2.5 Probe- Mm-Il7, GenBank: NM_008371.4 (2–1221), RNAscope 2.5 LS Probe- Mm-Tcrg-V6, GenBank: NG_007033.1: (2–475) and RNAscope LS 2.5 Probe- Mm-Trdc, GenBank: gi|372099096 (9–1098). Fluorescein staining was used as a dump channel for the exclusion of cells with unspecific background staining. Different combinations of fluorochromes were used for each probe to avoid bias in staining. Slides were scanned with Axio Scan (Zeiss), and images were analysed using Halo software (Indica Labs).

## *In vivo* cell proliferation assay

Every other day young and mid-aged mice were injected *i.p.* with 4 mg/kg of either IgG2b isotype control (clone MPC-11, BioXCell) or monoclonal antibody against IL-7 (clone M25, BioXCell; Fig 5I). Mice were given 40 mg/kg EdU through *i.p.* injection at day 1, and EdU was given in the drinking water at 0.5 mg/ml from day 1. Drinking water containing EdU was provided freshly every day, and the amount of water consumed by the mice was monitored. Mice were sacrificed at day 4, and immune cells were harvested from pLNs for EdU staining using the Click-iT Plus EdU pacific blue flow cytometry

kit (Thermo Fisher Scientific) along with antibodies against various cell surface markers (Appendix Table S1) for FACS analysis.

### *In vivo* tumour model

3LL-A9 cells were grown in DMEM (Gibco) supplemented with 10% FCS and tested negative for mycoplasma (MycoProbe® Mycoplasma Detection Kit, R&D systems) and mouse pathogens (M-LEVEL 1 analysis, Surrey Diagnostics). For injection, the right flank of young and old mice was shaved and $3 \times 10^6$ 3LL-A9 Lewis lung cancer cells were injected subcutaneously. Mice were sacrificed on day 14 or 15 postinoculation, and the tumour and tumour-draining LN were harvested for characterisation of γδ T cells by flow cytometry. FTY720 (Sigma-Aldrich) was reconstituted in ethanol and diluted in 2% β-cyclodextrin (Sigma-Aldrich) for injections. Mice were injected *i.p.* every other day with FTY720 at 1 mg/kg or with vehicle control three times before *s.c.* injection of $3 \times 10^6$ 3LL-A9 cells at right flank (Fig 6A). FTY720 treatment was continued on day 1 post-tumour cell injection for further seven times before tissue collection at day 14/15. Tumour tissue was weighed and minced by surgical curve scissors and then mashed through a 70-μm cell strainer (Greiner Bio-one) with the plunger of 2-ml syringe. Flow-through was passed again through a 40-μm cell strainer (Greiner bio-one) to prepare single-cell suspension. Immune cells were subsequently enriched from cell suspension by gradient centrifugation using Optiprep™ density gradient medium (Sigma-Aldrich). Briefly, cells were resuspended in 10 ml 33.3% Optiprep™ (diluted with PEB containing PBS with 0.5% BSA and 5 mM EDTA) and 5 ml PEB was layered gently on top of cell suspension without disturbing the interface of two layers. Cells were then centrifuged at $500 \times g$ for 20 min at 4°C without brake at the end of centrifugation. Immune cells at the interface between two layers were collected and washed twice with PBS before flow cytometric analysis.

### Statistical analysis

Statistical analysis was performed using Prism 7 software (GraphPad Inc.). Each data set was firstly analysed by D'Agostino and Pearson normality test for Gaussian distribution. Outliers were identified from each data set by ROUT test ($Q = 1\%$) and were excluded from subsequent analyses. Unpaired *t*-test was used for the comparisons between two data sets (young vs. old) both with a normal distribution. Comparisons between two groups (young vs. old) failed to pass normality test were performed using Mann–Whitney test. Two-way ANOVA with Sidak multiplicity correction test was used to compare multiple variables, such as γδ T-cell lineages and subsets, between two different groups (young vs. old). Descriptive statistics are expressed as mean ± SD (standard deviation) in all figures. All statistical analyses were performed as two-tailed tests, and the level of statistical significance in differences was indicated by *P*-values in all figures (\*$P < 0.05$; \*\*$P < 0.01$; \*\*\*$P < 0.001$; \*\*\*\*$P < 0.0001$).

### Statistical analysis for RNA-Seq data

Principal component analysis of normalised, $\log_{10}$-transformed counts was performed using the *prcomp* function in R. Differential expression analysis was performed using the Bioconductor package

*edgeR* [77]. A quasi-likelihood negative binomial generalised log-linear model was fitted to the count data after removing lowly expressed genes (averaged expression < 10 counts). The *glmQLFTest* function was used to perform genewise statistical testing incorporating the age of the animals as contrasts. Gene-level differential expression tests with a false discovery rate smaller than 10% were considered as statistically significant. To profile clonal diversity within each library, we calculated the inverse Simpsons index of the clone count as implemented in the R package *tcR* [78]. To allow the comparisons between libraries, we subsampled the clones to similar numbers within each T-cell subset.

## Data availability

The data sets produced in this study are available in the following databases:
RNA-Seq data: ArrayExpress E-MTAB-7178 (https://www.ebi.ac.uk/arrayexpress/experiments/E-MTAB-7178/).
Analysis scripts for the RNA-Seq and TCR analysis can be found at https://github.com/MarioniLab/GammaDeltaTcells2018.

**Expanded View** for this article is available online.

## Acknowledgements

Special thanks and gratitude go to Julia Jones and Cara Brodie from the Histopathology/ISH core at the CRUK Cambridge Institute for their support in conducting RNAscope experiments and analysis; Angela Mowbray, Matthew Clayton and Gemma Cronshaw from BRU for expert animal care and technical support; the Flow cytometry core for cell sorting; and genomics core for sequencing. We thank Prof. Adrian Hayday and Dr. Anett Jandke at King's College London, UK, for supplying 17D1 hybridoma supernatant and a FACS staining protocol. We thank Dr. Michelle Linterman (Babraham Institute, UK) for providing pLN samples of both young and aged mice from her ageing colony. We also thank Dr. Matthias Eberl (Cardiff University, UK) for constructive comments on the article. This work was supported by Cancer Research UK [MdlR (A22257), DTO (A22398), JCM (A22231); H-CC, CPM-J, VC]; Sir Henry Dale Fellowship jointly funded by the Wellcome Trust and the Royal Society [MdlR (WT107609); LMOB]; EMBL international PhD programme (NE); Janet Thornton Fellowship [CPM-J (WT098051)]; European Research Council [DTO (615584)]; EMBO Young Investigators Programme (DTO); EMBL (JCM); and the Wellcome Sanger Institute [JCM (105031/Z/14/Z); CPM-J, DTO].

## Author contributions

H-CC and MdlR designed the experiments; H-CC, CPM-J, LMOB and VC performed experiments and experimental analyses; NE performed computational analyses; DTO provided the ageing colony; JCM supervised computational analyses; H-CC and MdlR wrote the article. All authors commented on and approved the article.

### Conflict of interest

The authors declare that they have no conflict of interest.

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
