## [Review Process File · EMBO Reports]

IL-7-dependent compositional changes of the $\gamma\delta$ T cell pool in lymph nodes during ageing lead to an unbalanced anti-tumour response

Hung-Chang Chen, Nils Eling, Celia Pilar Martinez-Jimenez, Louise McNeill O'Brien, Valentina Carbonaro, John C. Marioni, Duncan T. Odom, and Maike de la Roche

Review timeline:

Submission date:	6 December 2018
Editorial Decision:	18 January 2019
Revision received:	3 May 2019
Editorial Decision:	17 May 2019
Revision received:	29 May 2019
Accepted:	5 June 2019

Editor: Achim Breiling

Transaction Report:

1st Editorial Decision

18 January 2019

Thank you for the submission of your research manuscript to EMBO reports. We have now received the full set of referee reports that are copied below.

As you will see, all three referees have major concerns regarding the publication of your manuscript in EMBO reports, although they acknowledge that the findings are of interest. However, the referees also state that data is largely descriptive, that causal relationships are not clear, and that the proposed mechanisms are not fully supported by the data. In particular, referees #2 and 3# indicate that a functional *in vivo* demonstration showing that (manipulating) IL-7 negatively affects IL17-producing $\gamma\delta$ T cells, that (manipulating) IL17-producing $\gamma\delta$ T cells affects tumor growth in old mice, would be needed for further consideration of the study in EMBO reports. As the reports are below, I will not further detail them here.

Given these comments, and considering the amount of work required to address them, we cannot offer to publish your manuscript. However, in case you feel that you can address the referee concerns in a timely and thorough manner, and can obtain data that would considerably strengthen the study as outlined in the referee reports, we would have no objection to consider a new manuscript on the same topic in the future. Please note that if you were to send a new manuscript this would be assessed again with respect to the literature and the novelty of your findings at the time of resubmission.

I am sorry to have to disappoint you this time. I nevertheless hope, that the referee comments will be helpful in your continued work in this area, and I thank you once more for your interest in our journal.

REFeree REPORTS

Referee #1:

Chen et al. report an original finding, namely the strong expansion of Vg6+ gdT17 cell in ageing mice. To my knowledge, this has not been reported as such before and will be of strong interest to the field. Furthermore, the authors propose a link between changes in the composition of gd T cells in the ageing lymph nodes with an increased risk of cancer development. The data look sound and convincing, and the authors employed a number of (single cell) sequencing technologies that are not standard in the gd T cell field. As ageing immune profiles are of broad interest to our ageing societies, these topics are also of general interest to the broad readership of EMBO. However, some issues should be addressed in a revised version:

Major:

A caveat is the emerging importance of the microbiome for the homeostasis of Vg6+ cells. This is not discussed in the manuscript and should be addressed experimentally. At least co-housing, swap of bedding between young and aged mice, or analysis of young versus middle-aged-versus old mice in different animal facilities should be performed to control the influence of the microbiome.

Along this line, what happens in the mesenteric lymph nodes in aged versus young mice? These are known to display fewer Vg6+ cells than pLN, however mLN-resident Vg6+ cells can expand and stay expanded in response to intestinal infection. (see also PMID 23890071)

Conceptually, the authors could clarify their rationale why increased numbers and frequencies of gd T17 cells lead to increased risk of cancer development in the elderly. gd T17 cells are viewed as pro-tumour, pro-inflammatory, and pro-angiogenic / this may contribute to, but is certainly not the only reason for age-associated immune functions. ... Without direct evidence, the authors should tone down title of their ms, i.e. that "altered composition enhances tumor growth".

In the TCR repertoire (Trd) analysis displayed in Fig. 4, the authors conclude a near exclusive usage of a Vd4 clone in Vg6+ gd T cells. This "Vd4" sequence is supposed to have an identical AA sequence as a previously described monoclonal Vd1 sequence pairing with Vg6 and Vg5 T cells. In the ms, this issue is discussed as in line with their ref # 47 and as controversial to previous descriptions of these almost monoclonal cells in their refs #42, 48, 49. In addition, a recent ms investigating Trd sequences of expanded Vg6+ revealed the same CDR3 as a Vd1 sequence (PMID 27851927). maybe there is/ was a bug in IMGT? For a molecular biology focused journal, this issue should be clarified in detail before publication. Eventually, commercially available mAb specific for Vd4 could be employed.

Minor:

If statistics reveal no significant difference between tested groups, this should be indicated as n.s. in the respective figure (e.g. Fig. 1E)

Fig. 5D is appearing directly after Fig 1B, before 1C. Not sure about EMBO's policy on figure call out, should be sequential?

In my personal view, upper case "-ve" for "negative" is not easy to grasp for the readers.

Inconsistent AA sequence of canonical Vg4/Vd5 clone in text (line 256) and Fig. 4C line 84, "upon tumour challenge, Vg6+ gdT17 cells become activated in pLNs, migrate into the tumour and create a pro-tumour microenvironment ..." - this is not restricted to aged mice, correct? please adjust.

There are a couple of groups that investigate the impact of ageing on gd T cell clonotypes phenotypes and overall numbers mostly human peripheral blood, e.g. the work of Langerak et al. in PMID 29559980 and 28710491 could be discussed.

line 217: repertoire diversity?

line 357: please show also Vg6+ gd T cells from non-draining pLN for comparison to make this statement.

Referee #2:

The manuscript by Chen et al. entitled "Altered composition of the $\gamma\delta$ T cell pool in lymph nodes during aging enhances tumor growth" reports on age-associated changes in $\gamma\delta$ T cells. The authors find that IL-17-producing $\gamma\delta$ T cells expand in the lymph nodes of old mice (>21 months old), when compared to young mice (3 months old) and these cells express the V γ 6 T cell receptor. Interestingly, the gene expression profile of $\gamma\delta$ T cells from young and old mice show negligible differences, but the T cell receptor repertoire exhibits some differences between young and old mice. The authors found that expression of IL-7 and IL-2 increase in lymph nodes of old mice and IL-7 expression localizes to regions where IL-17-producing $\gamma\delta$ T cells reside in lymph nodes. The authors used the inhibitor FTY720 to block T cell egress from the lymph node and observed less T cells in tumors of these mice. The authors show that injecting the same cancer cell line, a Lewis lung carcinoma variant, subcutaneously into young or old mice results in greater tumor weight in old mice. These tumors from old mice contained more IL-17-producing $\gamma\delta$ T cells than tumors from young mice.

Overall, this study represents a thorough characterization of $\gamma\delta$ T cells in aging mice. The experiments are nicely controlled with adequate power, and the statistics are correctly performed. The authors rightly point out the novelty of study and the data will be an important resource for the $\gamma\delta$ T cell community. The drawback to the study is that it is largely descriptive with mostly correlative data - only one intervention experiment was performed ($\gamma\delta$ T cell egress from lymph nodes using FTY720). Therefore, the statement in the Abstract, "...changes of $\gamma\delta$ T cell pool...promote tumor growth" and similar statements in the Introduction and Discussion are not supported by experimental evidence. The authors fail to provide a causal relationship between increased IL-7 expression in lymph nodes and expansion of IL-17-producing $\gamma\delta$ T cells in old mice, together with increased tumor growth.

Below are some minor comments that should be addressed to strengthen the manuscript:

1. I disagree with the conclusion made from Figure 1B. If there is no statistical difference between the groups, then the absolute number of $\gamma\delta$ T cells in lymph nodes is the same in young and old mice.
2. On page 4, the authors state that there was no association between obesity and $\gamma\delta$ 17 phenotype. Please provide the data.
3. In Figure 6A, what subset of $\gamma\delta$ T cells are affected by the FTY720 treatment?

Referee #3:

Ageing is associated with a decline of conventional T cell response that is known to influence susceptibility to infection and incidence of cancer. However, the effect of ageing on the biology of innate-like gd T cells remains little elucidated. In this manuscript Chen et al analysed how ageing affects the gd T cell subsets and their functions. Remarkably, the proportion of IL-17-producing gd T cells increased at steady state in old mice constituting the majority of the gd T cells (over IFN-g-producing gd T cells) present in lymph nodes and to a lesser extent in spleen. Evidence supports a mechanism extrinsic to gd T cells that depends on increase in IL-7 production in the T zone of the lymphoid organs. The main IL-17+ gd T cell subsets that increase in proportion are Vg6 and Vg4 gd T cells. The manuscript is well presented and written. The data at steady state are overall convincing and most interpretations are adequate. However, in its present form the work relies exclusively on observations and lacks correlations to support the proposed mechanisms. For instance while the changes in the proportion of IL-17+ gd T cells at steady state in old mice are convincing the physiological impact of the presence of these IL-17+ gd T cells on tumour growth remains to be better characterised. Therefore, the manuscript would substantially benefit from direct in vivo functional demonstration that i) IL-7 supports a niche rich in IL-17+ gd T cells in lymph nodes, and ii) that the latter have a major impact on tumour growth.

Major Concerns:

- 1) P4 L148 > "The prevalent IFN-g response by gd T cells in young mice becomes skewed towards an IL-17-dominated response during ageing." It is unclear as to whether the total IL-17 production is

increased in lymph nodes of old mice compared to young mice, or if it is simply the proportion of IL-17+ gd T cell subsets within the total gd T cells. This could be tested by assessing the expression of IL-17 by intracellular flow cytometry staining from the CD3+ population after PMA-Iono restimulation, and by assessing IL-17 mRNA from total cell suspension by real time-PCR, both from lymph nodes from young and old mice.

In the same way for figure 6, it would be important to show that the levels of IL-17 (protein and/ or mRNA) are higher in tumours harvested from old mice compared to young mice.

2) P7 L300 > "Taken together, we show that IL-7 production in the T cell zone of pLNs is highly increased upon ageing and correlates with the expansion of gd17 T cells, especially Vg6+ T cells, resulting in a skewed peripheral gd T cell pool that might favour pro-inflammatory immune responses." The increase level of Il7 mRNA expression in the T cell zone of lymph node, at best, associates with the accumulation of gd17 T cells. Therefore it is suggested to demonstrate the possible correlation by showing that systemic injection of an antagonist anti-IL-7 antibody would lead to a decrease in gd17 T cells in old mice (which would maintain similar levels as to young mice).

3) P7 L318 > "Strikingly, we found that 3LL-A9 tumours developed faster in old mice (Fig. 6C)." To support that tumours grow faster in old compared to young mice, it would be informative to plot the tumour growth (X= days; Y= tumour size (mm³)) by calculating the tumour volume with H and L (as measured with a calliper). The approximation of the tumour volume is then given with the following formula: (L x H x H)/2 and plotted as a function of time (days). Visualisation of the kinetics of tumour growth will be more convincing than only one measurement given in grams 14 days after transplantation of the tumour.

4) P1 abstract L29 > "Importantly, IL-17-producing gd17 T cells dominated the gd T cell pool of aged mice - mainly due to the selective expansion of Vg6+ gd17 T cells and augmented gd17-polarisation of Vg4+ T cells." The mechanism(s) leading to increased proportion and number of IL-17-producing gd17 T cells is unclear. "Expansion" suggests proliferation but it is not demonstrated, while "augmentation" does not distinguish between proliferation, recruitment or cell death of the other subsets. Thus, it would be appropriate to use augmentation instead of expansion. Then, it would be interesting to clarify the mechanism of augmentation of Vg6+ gd17 T cells and gd17-polarisation of Vg4+ T cells. To do so, it is recommended to perform an experiment of BrdU incorporation, or Ki67 staining and caspase 3 staining of gd17 T cells harvested from mice at 12 months and 21 months.

5) Evidence has emerged of a crosstalk between gd17 T cells and neutrophils in various contexts (such as cancer and infections) (Cheemarla et al., 2017; Liu et al., 2016; Mensurado et al., 2018; Wozniak et al., 2012). In particular neutrophils have been shown in tumours to limit the presence of gd17 T cells in a way that affects tumour growth (Liu et al., 2016; Mensurado et al., 2018). Thus, beside a role of IL-7 on creating a favourable niche for gd17 T cells in the lymph nodes, it would be interesting to determine a potential role for neutrophils in tumours of old mice. To do so assessment of neutrophils (CD45+ live/dead-ve CD11b+ Ly6Cint Ly6GHi) by flow cytometry should be performed in tumours harvested from young and old mice.

6) P8 L360 > "Furthermore, the biased gd17 T cell pool in pLNs of old mice correlates with a pro-tumour microenvironment and enhanced tumour progression." While the existence of a gd17 T cell bias with age is convincing, its physiologic relevance still remains to be clearly established. To establish clearly a correlative link between gd17 T cells and tumour progression, it would be critical to assess the tumour growth in young and old mice deficient in IL-17 (IL-17^{-/-} mice) and gd T cells (TCRd^{-/-} mice). As an alternative, injection of blocking antibodies specific for mouse IL-17 and TCRd (such as GL3, and GL4 (Goodman and Lefrancois, 1989; Koenecke et al., 2009; Sheridan et al., 2013)) should be considered and followed by direct measurements of tumour growth. Of note: it was shown that the GL3 antibody induces a state of anergy of the gd T cells due to TCR internalisation, instead of being depleted (Koenecke et al., 2009), nevertheless this approach should impair IL-17+ invariant Vg6 T cells that can recognise tumour-associated antigens or other signals, at least in the 3LL-A9 model (see P10 L415).

Minor Issues:

1) There is a typo in figure 5E on the Y axis "% Vg6+ ceels in all T cells" should be changed to "%

Vg6+ cells in all T cells".

2) Please refrain from using "tumour development" because tumours only develop in spontaneous genetic mouse models of cancer (such as mice with mutations in p53 and Brca1 (breast) or Braf and Pten (skin) for example...). When tumour cells are injected subcutaneously, that is implanted, the development phase is bypassed and we should refer to tumour growth, progression or assessment of tumour burden.

3) P7 L300 > "Taken together, we show that IL-7 production in the T cell zone of pLNs is highly increased upon ageing and correlates with the expansion of gd17 T cells, especially Vg6+ T cells, resulting in a skewed peripheral gd T cell pool"

P8 L360 > "Furthermore, the biased gd17 T cell pool in pLNs of old mice correlates with a pro-tumour microenvironment and enhanced tumour progression."

As indicated above, the present work does not provide clear demonstration that IL-7 is indeed the extrinsic cytokine in the microenvironment that promotes the accumulation of gd17 T cells. In the same way, there is no evidence that the presence of gd17 T cells indeed supports tumour progression. There is at best an association, but not a correlation, thus please only use "correlation" in the few appropriate contexts throughout the manuscript.

References cited:

Cheemarla, N.R., Baños-Lara, M.D.R., Naidu, S., and Guerrero-Plata, A. (2017). Neutrophils regulate the lung inflammatory response via $\gamma\delta$ T cell infiltration in an experimental mouse model of human metapneumovirus infection. *J. Leukoc. Biol.* 101, jlb.4A1216-519RR.

Goodman, T., and Lefrancois, L. (1989). Intraepithelial lymphocytes. Anatomical site, not T cell receptor form, dictates phenotype and function. *J. Exp. Med.* 170, 1569-1581.

Koenecke, C., Chennupati, V., Schmitz, S., Malissen, B., Förster, R., and Prinz, I. (2009). In vivo application of mAb directed against the gammadelta TCR does not deplete but generates "invisible" gammadelta T cells. *Eur. J. Immunol.* 39, 372-379.

Liu, Y., O'Leary, C.E., Wang, L.-C.S., Bhatti, T.R., Dai, N., Kapoor, V., Liu, P., Mei, J., Guo, L., Oliver, P.M., et al. (2016). CD11b+Ly6G+ cells inhibit tumor growth by suppressing IL-17 production at early stages of tumorigenesis. *Oncoimmunology* 5, e1061175.

Mensurado, S., Rei, M., Lanca, T., Ioannou, M., Gonçalves-Sousa, N., Kubo, H., Malissen, M., Papayannopoulos, V., Serre, K., and Silva-Santos, B. (2018). Tumor-associated neutrophils suppress pro-tumoral IL-17+ $\gamma\delta$ T cells 1 through induction of oxidative stress. *PLoS Biol* 16, 1-21.

Sheridan, B.S., Romagnoli, P. a, Pham, Q.-M., Fu, H.-H., Alonzo, F., Schubert, W.-D., Freitag, N.E., and Lefrançois, L. (2013). $\gamma\delta$ T cells exhibit multifunctional and protective memory in intestinal tissues. *Immunity* 39, 184-195.

Wozniak, K.L., Kolls, J.K., and Wormley, F.L. (2012). Depletion of neutrophils in a protective model of pulmonary cryptococcosis results in increased IL-17A production by $\gamma\delta$ T cells. *BMC Immunol.* 13, 65.

1st Revision - authors' response

3 May 2019

We are re-submitting our manuscript "Altered composition of the gd T cell pool in lymph nodes during ageing enhances tumour growth" (tracking number EMBOR-2018-47379V1), the title of which we have changes to "IL-7-dependent compositional changes of the gd T cell pool in lymph nodes during ageing lead to an unbalanced anti-tumour response" in response to the referee comments. We have performed additional experiments that have substantially improved the manuscript and have addressed all comments from the three reviewers. A detailed point by point response to each referee can be found below.

In the revised manuscript we have confirmed that the selective expansion of Vg6+gd17 T cells upon ageing is conserved not only in the peripheral lymph nodes (pLNs) but also in mesenteric LNs and spleen (*new. Suppl. Fig. 4*) and is independent of the local microbiome (*new. Suppl. Fig. 3*). Furthermore, we have shown that the expanded gd17 T cells produce the majority of IL-17 in pLNs of old mice (*new. Suppl. Fig. 2 F&H*).

Most importantly, we have performed functional intervention experiments by neutralizing IL-7 *in vivo* (*new Fig. 5I*). Strikingly, we find that the proliferation of gd17 T cells, in particular Vg6+ gd17 T cells, but not gd1 T cells in the pLN is mediated by IL-7 (*new Fig. 5J and K, new Suppl. Fig. 9, new Suppl. Fig. 10*), thus providing the mechanistic insight into the expansion of Vg6+ gd17 T cells upon ageing. The results are nicely in line with our previous results showing that IL-7 is highly expressed in the aged LN, confirming that gd17 T cells express higher levels of IL-7Ra compared with gd1 T cells, and demonstrating that Vg6+ gd17 T cells are localised in close proximity to IL-7 producing cells. Taken together, we propose that IL-7-producing cells in the pLNs create a niche for local Vg6+ gd17 T cell expansion.

We show that the pro-tumour Vg6+ gd17 T cells are selectively activated in the pLN, migrate into the tumour and represent the only activated and exhausted gd T cell subset in the tumour microenvironment (TME). We also show that Vg6+ gd17 T cells are the principle IL-17-producing T cells in the tumour (*new. Suppl. Fig. 4*) and have excluded differential infiltration of neutrophils as a factor affecting tumour growth (*new. Fig. 6I*).

The reviewers suggested strengthening the functional link between increased pro-tumour Vg6+ gd17 T cells in old mice and enhanced tumour growth by ablating gd T cells, but unfortunately there are no reagents available to specifically ablate Vg6+ gd17 T cells. In an attempt to address this issue, we have injected old mice with pan-gdTCR antibody (Figure for reviewers, at the end of the point by point response) despite the fact that treatment with pan-gdTCR antibodies has two major limitations. First, administration of pan-TCRgd antibodies into mice does not deplete gd T cells but results in internalisation of the gdTCR by gd T cells, rendering the cells undetectable while resulting in their activation (Koenecke *et al.*, 2009). Second, the treatment with pan-TCRgd antibodies affects both gd17 and gd1 T cells and the presence of gd1 T cells in the TME could be important for inhibiting tumour growth. The outcome of these experiments was inconclusive, unsurprisingly in view with these limitations, as we found very heterogeneous responses in pan-gdTCR antibody treated mice. Because of these technical limitations, which preclude a rigorous interpretation of the results, we have provided the results for the reviewers only but have chosen not to include them in the manuscript.

We believe that our manuscript is an excellent fit for EMBO Reports and are delighted that the reviewers agree that our findings are cutting edge, experimentally sound, and of great relevance not only to the gd T cell and ageing fields but also to the broader scientific community.

Referee #1:

Chen et al. report an original finding, namely the strong expansion of V6+ gdT17 cell in ageing mice. To my knowledge, this has not been reported as such before and will be of strong interest to the field. Furthermore, the authors propose a link between changes in the composition of gd T cells in the ageing lymph nodes with an increased risk of cancer development. The data look sound and convincing, and the authors employed a number of (single cell) sequencing technologies that are not standard in the gd T cell field. As ageing immune profiles are of broad interest to our ageing societies, these topics are also of general interest to the broad readership of EMBO. However, some issues should be addressed in a revised version:

Major:

A caveat is the emerging importance of the microbiome for the homeostasis of Vg6+ cells. This is not discussed in the manuscript and should be addressed experimentally. At least co-housing, swap of bedding between young and aged mice, or analysis of young versus middle-aged-versus old mice in different animal facilities should be performed to control the influence of the microbiome.

*> Recently, the local microbiome of lung and gingiva has emerged to play an important role in the homeostasis of Vg6+ T cells in these tissues (Wilharm *et al.*, 2019 and Jin *et al.*, 2019). We have analysed the gd T cell pool of peripheral lymph nodes (pLNs) from both, young and old C57BL/6 mice, bred and housed in a different animal facility to control for a potential influence of the local*

microbiome of the skin, lung or digestive tract in our mouse facility on the expansion of Vg6+ gd17 T cells in the pLNs upon ageing.

From the healthy ageing cohort at the Babraham Institute (UK), we were able to obtain pLNs from young (3 month) and old (21 month) mice that had been previously vaccinated with a mixture of cholera toxin (CTx) and CTx-NP peptide conjugates by oral gavage. There are no reports in the literature suggesting that this treatment affects the local microbiomes of the animals or the composition of the pLNs. However, transient immune activation in the gut and systemic antibody responses are observed after oral gavage of CTx or CTx-NP. We have added this information into the Materials and Methods section.

From the Babraham cohort, we have obtained results that are identical to our previous findings. We have included these results in the *new Suppl. Fig. 3*, arguing that the expansion of Vg6+ gd17 T cells in the pLNs upon ageing occurs irrespective of the local microbiomes, transient immune activation in the gut and systemic antibody responses.

Of note, "ageing" is now a regulated procedure under UK Home Office regulation and aged mice cannot be obtained easily from a lot of mouse facilities since mice >12month of age have to be killed.

Along this line, what happens in the mesenteric lymph nodes in aged versus young mice? These are known to display fewer Vg6+ cells than pLN, however mLN-resident Vg6+ cells can expand and stay expanded in response to intestinal infection. (see also PMID 23890071)

> We have analysed the mesenteric lymph nodes (mLNs) of young and old mice from our healthy C57BL/6 colony housed in a clean animal facility under SPF conditions negative for murine intestinal pathogens. We confirmed that young mice have fewer Vg6+ T cells in the mLN compared with the pLNs. Similar to what we have observed in the spleen (Suppl. Fig. 5) we found an increase in gd17 T cells and a decline of the gd1 lineage upon ageing, albeit to a lesser degree compared with pLNs. The proportion of Vg6+ T cells was also significantly increased and Vg1+ T cells were decreased in mLN from old mice with the changes again being less severe compared with pLNs. However, the ~2-fold increase in the proportion of gd T cells in the T cell pool that was observed in the pLN was not seen in the mLN.

The data obtained is presented in the *new Suppl. Fig. 4*.

Conceptually, the authors could clarify their rationale why increased numbers and frequencies of gd T17 cells lead to increased risk of cancer development in the elderly. gd T17 cells are viewed as pro-tumour, pro-inflammatory, and pro-angiogenic / this may contribute to, but is certainly not the only reason for age-associated immune functions. ... Without direct evidence, the authors should tone down title of their ms, i.e. that "altered composition enhances tumor growth".

> In our manuscript, we have now proven functionally that expansion of the gd17 T cells, in particular Vg6+ T cells, in the pLN is driven by IL-7 *in vivo* (*new Fig. 5I, new Suppl. Fig. 9, and new Suppl. Fig. 10*). We have shown for the first time that gd T cells from the LN make up 80% of the gd T cell pool of the tumour by blocking egress of gd T cells *in vivo* with FTY720 treatment (Fig. 6A). Furthermore, we found that Vg6+ gd17 T cells become solely activated in the tumour-draining LN (Suppl. Fig. 14) and are the dominating gd T cell subset in tumours of old mice (Fig. 6E). Due to the well-reported pro-tumour activities of gd17 T cells, these Vg6+ gd17 T cells are excellent candidates for generating a pro-tumour environment leading to enhanced tumour growth especially in old mice. And we show that their presence in the tumour microenvironment (TME) strongly correlates with tumour growth (Fig. 6F).

Direct functional evidence for the protumour role of Vg6+ gd17 T cells in our model could be obtained from experiments specifically blocking or neutralising Vg6+ T cells *in vivo*. There are no reagents available to do this. We have attempted to functionally deplete all gd T cells in old mice with a pan-TCRgd antibody (Figure for reviewers), but this experiment is uninterpretable for two reasons. First, administration of pan-TCRgd antibodies into mice does not deplete gd T cells but results in internalisation of the gdTCR by gd T cells, rendering the cells undetectable while also leading to activation of these "invisible" cells (Koenecke *et al.*, 2009). Second, the treatment with pan-TCRgd antibodies affects both gd17 and gd1 T cells and the presence of gd1 T cells in the TME could be important for inhibiting tumour growth. Thus, we cannot really conclude anything from the fact that pan-gd T cell depletion did not affect tumour growth nor infiltration of various immune cell subsets, and have chosen not to include it in the study (but present it for the reviewer).

We also acknowledge that they are likely additional mechanisms, *e.g.* reduced cytotoxic activity of CD8+ T cells that can promote tumour growth in old mice and have thus toned down the title of our manuscript.

The title reads now: “IL-7-dependent compositional changes of the gd T cell pool in lymph nodes during ageing lead to an unbalanced anti-tumour response”.

We pointed out correlations where we failed to provide clear functional evidence due to the lack of suitable reagents.

P1 L36: Thus, upon ageing, substantial compositional changes of gd T cell pool in the pLN lead to an unbalanced gd T cell response in the tumour that is associated with accelerated tumour growth.

P2 L87: Upon tumour challenge, Vg6+ gd17 T cells become activated in pLNs, migrate into the tumour and create a pro-tumour microenvironment that is associated with enhanced tumour growth.

P10 L416: The biased gd17 T cell pool in pLNs under homeostatic condition upon ageing augments the infiltration of pro-tumour Vg6+ gd T cells, which is associated with enhanced tumour growth in aged mice.

P11 L597: Taken together, we have identified a novel age-dependent dysregulation of the gd T cell pool that is associated with enhanced tumour progression in old mice.

In the TCR repertoire (Trd) analysis displayed in Fig. 4, the authors conclude a near exclusive usage of a Vd4 clone in Vg6+ gd T cells. This "Vd4" sequence is supposed to have an identical AA sequence as a previously described monoclonal Vd1 sequence pairing with Vg6 and Vg5 T cells. In the ms, this issue is discussed as in line with their ref # 47 and as controversial to previous descriptions of these almost monoclonal cells in their refs #42, 48, 49. In addition, a recent ms investigating Trd sequences of expanded Vg6+ revealed the same CDR3 as a Vd1 sequence (PMID 27851927). maybe there is/ was a bug in IMGT? For a molecular biology focused journal, this issue should be clarified in detail before publication. Eventually, commercially available mAb specific for Vd4 could be employed.

> We have resolved our controversial results on the pairing of Vg6 chain with Vd4 chain instead of Vd1 chain as previously reported. In the previous manuscript, we wrongly allocated TCRDV4 (IMGT gene name) to the Vd4 chain instead of Vd1. In the revised manuscript, we have corrected the annotation of different TRDV genes to the corresponding Vd chain as designated in the literature.

We have corrected Fig. 4 in manuscript and the text accordingly and are now showing delta chains by IMGT gene subgroup with the designated Vd chain in brackets.

For clarity, we have included the *new Suppl. Table 2* cross-referencing the different TCRDV gene names with actual Vd chain to avoid confusion.

Minor:

If statistics reveal no significant difference between tested groups, this should be indicated as n.s. in the respective figure (e.g. Fig. 1E)

> For clarity of the figures we point out statistically significant differences only.

Fig. 5D is appearing directly after Fig 1B, before 1C. Not sure about EMBO's policy on figure call out, should be sequential?

> We have omitted the discussion of Fig. 5D from *P3 L103* and in the revised manuscript discuss the figure in sequential order on *P7 L598*.

In my personal view, upper case "-ve" for "negative" is not easy to grasp for the readers.

> We have replaced all upper case “-ve” throughout the manuscript and figures with upper case “neg” for better clarity.

Inconsistent AA sequence of canonical Vg4/Vd5 clone in text (line 256) and Fig. 4C

> The inconsistent AA sequence is explained by the different output formats of IMGT and MiXCR. MiXCR alignment output contains one additional amino acid in the front and at the rear of the IMGT sequence (CASGYIGGIRATDKLVF). We have now synchronised the presentation of this AA sequence in the manuscript and in Fig. 4C.

line 84, "upon tumour challenge, Vg6+ gdT17 cells become activated in pLNs, migrate into the tumour and create a pro-tumour microenvironment ..." - this is not restricted to aged mice, correct? please adjust.

> The statement holds true in both young and old mice and we have deleted “in aged mice” from *P2 L87* to reflect this.

There are a couple of groups that investigate the impact of ageing on gd T cell clonotypes phenotypes and overall numbers mostly human peripheral blood, e.g. the work of Langerak et al. in PMID 29559980 and 28710491 could be discussed.

> Unfortunately, gd T cell subsets in humans cannot be directly compared with the ones in mouse, and gd T cells in the peripheral blood are distinct from the ones in the peripheral lymphoid organs. However, we find the published work on the effect of ageing on human gd T cells in the blood very interesting and have included a paragraph in the discussion emphasising the conceptually common findings (P11 L491).

“Interestingly, a shift towards more effector phenotypes, largely maintained TCR diversity and a change in Vg/Vd usage have also been detected in the peripheral blood of humans upon organismal ageing [72, 73] suggesting similar age-related processes occurring in the murine and human gd T cell pool.”

line 217: repertoire diversity?

> We have corrected this typo in the revised manuscript.

line 357: please show also Vg6+ gd T cells from non-draining pLN for comparison to make this statement.

> In order to avoid any impact on mobility of our experimental mice, we injected the tumour cells very close to the back/spine of the mice. Unfortunately, this has resulted in both inguinal LNs on the left and right flank “draining” the tumour.

To support our statement, we have included the *new Suppl. Fig. 15* and show the activation status of gd T cells in the pLNs of young and old mice under homeostatic condition. No expression of Tim-3 and PD-1 by gd T cells was detected.

Referee #2:

The manuscript by Chen et al. entitled "Altered composition of the $\gamma\delta$ T cell pool in lymph nodes during aging enhances tumor growth" reports on age-associated changes in $\gamma\delta$ T cells. The authors find that IL-17-producing $\gamma\delta$ T cells expand in the lymph nodes of old mice (>21 months old), when compared to young mice (3 months old) and these cells express the V γ 6 T cell receptor.

Interestingly, the gene expression profile of $\gamma\delta$ T cells from young and old mice show negligible differences, but the T cell receptor repertoire exhibits some differences between young and old mice. The authors found that expression of IL-7 and IL-2 increase in lymph nodes of old mice and IL-7 expression localizes to regions where IL-17-producing $\gamma\delta$ T cells reside in lymph nodes. The authors used the inhibitor FTY720 to block T cell egress from the lymph node and observed less T cells in tumors of these mice. The authors show that injecting the same cancer cell line, a Lewis lung carcinoma variant, subcutaneously into young or old mice results in greater tumor weight in old mice. These tumors from old mice contained more IL-17-producing $\gamma\delta$ T cells than tumors from young mice. Overall, this study represents a thorough characterization of $\gamma\delta$ T cells in aging mice. The experiments are nicely controlled with adequate power, and the statistics are correctly performed. The authors rightly point out the novelty of study and the data will be an important resource for the $\gamma\delta$ T cell community. The drawback to the study is that it is largely descriptive with mostly correlative data - only one intervention experiment was performed ($\gamma\delta$ T cell egress from lymph nodes using FTY720). Therefore, the statement in the Abstract, "...changes of $\gamma\delta$ T cell pool...promote tumor growth" and similar statements in the Introduction and Discussion are not supported by experimental evidence. The authors fail to provide a causal relationship between increased IL-7 expression in lymph nodes and expansion of IL-17-producing $\gamma\delta$ T cells in old mice, together with increased tumor growth.

> In our revised manuscript we have now added a substantial amount of functional data. By neutralizing IL-7 *in vivo* in combination with EdU labelling we were able to show that the expansion of gd17 T cells, in particular Vg6+ T cells, in the pLNs is mediated by IL-7 (*new Fig. 5J and K, new Suppl. Fig. 9, and new Suppl. Fig. 10*).

By blocking egress from the LNs *in vivo* with FTY70, we have shown that 80% of the gd T cells in the tumour are coming from the pLNs (Fig. 6A). In addition, we show that Vg6+ gd17 T cells become solely activated in the tumour-draining LN (Suppl. Fig. 14) and are the dominating gd T cell subset in tumours of old mice (Fig. 6E). They produce the majority of IL-17 in the tumour (*new Suppl. Fig. 13*) and their performance is not preferentially inhibited by neutrophils that can block

pro-tumour gd17 function (*new Fig. 6I*). Due to the well-reported pro-tumour activities of gd17 T cells these Vg6+ gd17 T cells are excellent candidates for generating a pro-tumour environment especially in old mice. In addition, we show that their presence in the tumour microenvironment strongly correlates with tumour growth (Fig.6 F).

Direct functional evidence for the protumour role of Vg6+ gd17 T cells in our model could be obtained from experiments specifically blocking or neutralising Vg6+ T cells *in vivo*. There are no reagents available to do this. We have attempted to functionally deplete all gd T cells in old mice with a pan-TCRgd antibody but failed to demonstrate an effect on tumour growth and infiltration of various immune cell subsets (Figure for reviewers). Pan-gd T cell depletion is problematic for two main reasons. First, administration of pan-TCRgd antibodies into mice does not deplete gd T cells but leads to internalisation of the gdTCR rendering the cells undetectable while leading to activation of the “invisible” cells (Koenecke *et al.*, 2009). Second, the treatment with pan-TCRgd antibodies affects both - gd17 and gd1 T cells - and the presence of gd1 T cells in the TME could be important for inhibiting tumour growth.

We have changed the title of our manuscript (now: “IL-7-dependent compositional changes of the gd T cell pool in lymph nodes during ageing lead to an unbalanced anti-tumour response”), and pointed out correlations where we failed to provide clear functional evidence due to the lack of suitable reagents.

P1 L36: Thus, upon ageing, substantial compositional changes of gd T cell pool in the pLN lead to an unbalanced gd T cell response in the tumour that is associated with accelerated tumour growth.

P2 L87: Upon tumour challenge, Vg6+ gd17 T cells become activated in pLNs, migrate into the tumour and create a pro-tumour microenvironment that is associated with enhanced tumour growth.

P10 L416: The biased gd17 T cell pool in pLNs under homeostatic condition upon ageing augments the infiltration of pro-tumour Vg6+ gd T cells, which is associated with enhanced tumour growth in aged mice.

P11 L597: Taken together, we have identified a novel age-dependent dysregulation of the gd T cell pool that is associated with enhanced tumour progression in old mice.

Below are some minor comments that should be addressed to strengthen the manuscript:

1. I disagree with the conclusion made from Figure 1B. If there is no statistical difference between the groups, then the absolute number of $\gamma\delta$ T cells in lymph nodes is the same in young and old mice.
> The difference between young and old mice is statistically significant after removal of outliers by ROUT test (Q = 1%) (GraphPad PRISM). We have revised Fig. 1B and added this statement in the revised manuscript.

2. On page 4, the authors state that there was no association between obesity and $\gamma\delta 17$ phenotype. Please provide the data.

> Every mouse analysed was visually examined and around 40% of the aged mice were identified as obese. We observed no statistically significant difference in the proportion of gd17 T cells within pLN pool between normal and obese aged mice. We thus conclude that obesity has no confounding effect on the gd17 bias in the periphery. This observation is in line with recent work showing that obese (*ob/ob*) mice on a normal diet do not have increased numbers of gd17 T cells (Nakamizo *et al.*, 2017). The result is included in the revised manuscript as a *new panel in Suppl. Fig 2 (Suppl. Fig. 2E)*.

3. In Figure 6A, what subset of $\gamma\delta$ T cells are affected by the FTY720 treatment?

> We have included a *new panel in Suppl. Fig. 11 (Suppl. Fig. C and D)* to show the results. We found that the composition of the gd T cell pool was altered by FTY720 treatment. Progenitor gd T cells increased while gd1 T cells decreased upon FTY720. On the gd T cell subset level, Vg1+ and Vg6+ gd T cells declined, and Vg2/3/7+ and Vg4+ gd T cells increased, suggesting different levels of egress from the pLNs.

Referee #3:

Ageing is associated with a decline of conventional T cell response that is known to influence susceptibility to infection and incidence of cancer. However, the effect of ageing on the biology of

innate-like gd T cells remains little elucidated. In this manuscript Chen et al analysed how ageing affects the gd T cell subsets and their functions. Remarkably, the proportion of IL-17-producing gd T cells increased at steady state in old mice constituting the majority of the gd T cells (over IFN-g-producing gd T cells) present in lymph nodes and to a lesser extent in spleen. Evidence supports a mechanism extrinsic to gd T cells that depends on increase in IL-7 production in the T zone of the lymphoid organs. The main IL-17+ gd T cell subsets that increase in proportion are Vg6 and Vg4 gd T cells. The manuscript is well presented and written. The data at steady state are overall convincing and most interpretations are adequate. However, in its present form the work relies exclusively on observations and lacks correlations to support the proposed mechanisms. For instance while the changes in the proportion of IL-17+ gd T cells at steady state in old mice are convincing the physiological impact of the presence of these IL-17+ gd T cells on tumour growth remains to be better characterised. Therefore, the manuscript would substantially benefit from direct in vivo functional demonstration that i) IL-7 supports a niche rich in IL-17+ gd T cells in lymph nodes, and ii) that the latter have a major impact on tumour growth.

Major Concerns: 1) P4 L148 > "The prevalent IFN-g response by gd T cells in young mice becomes skewed towards an IL-17-dominated response during ageing." It is unclear as to whether the total IL-17 production is increased in lymph nodes of old mice compared to young mice, or if it is simply the proportion of IL-17+ gd T cell subsets within the total gd T cells. This could be tested by assessing the expression of IL-17 by intracellular flow cytometry staining from the CD3+ population after PMA-Iono restimulation, and by assessing IL-17 mRNA from total cell suspension by real time-PCR, both from lymph nodes from young and old mice. In the same way for figure 6, it would be important to show that the levels of IL-17 (protein and/ or mRNA) are higher in tumours harvested from old mice compared to young mice.

> As suggested by the reviewer, we have assessed the expression of IL-17 by different subsets of T cells from the pLNs of young and old mice upon *ex vivo* restimulation with PMA/ionomycin, and included the results in the *new Suppl. Fig. 2F and H*.

The proportion of IL-17-producing CD3+ T cells is increased six-fold in the pLNs of old mice (Suppl. Fig. 2F). gd T cells represent 40% and over 50% of the IL-17-producing CD3+ T cells in pLNs of young and old mice, respectively (Fig. 1E in manuscript). Memory CD4+ T cells accounted for the remaining IL-17 production in the pLNs. However, only half of the old mice showed an increase in IL17+ memory CD4+ T cells (Suppl. Fig.2H), making the increase of gd17 T cells the primary cause of the greatly increased IL17 production in pLNs of old mice.

In the same way we have analysed IL-17 production by T cells from the tumour of old and young mice upon restimulation *ex vivo* (*new Suppl. Fig. 13*). On average 70% of all IL-17 produced by T cells in the tumour is from gd T cells. Surprisingly, Vg6+ gd17 T cells from old mice produced less IL-17 upon restimulation. We think this is likely caused by the high level of exhaustion due to the early activation and tumour cell infiltration of these cells in old mice.

Point 2) & 4)

2) P7 L300 > "Taken together, we show that IL-7 production in the T cell zone of pLNs is highly increased upon ageing and correlates with the expansion of gd17 T cells, especially Vg6+ T cells, resulting in a skewed peripheral gd T cell pool that might favour pro-inflammatory immune responses." The increase level of Il7 mRNA expression in the T cell zone of lymph node, at best, associates with the accumulation of gd17 T cells. Therefore, it is suggested to demonstrate the possible correlation by showing that systemic injection of an antagonist anti-IL-7 antibody would lead to a decrease in gd17 T cells in old mice (which would maintain similar levels as to young mice).

4) P1 abstract L29 > "Importantly, IL-17-producing gd17 T cells dominated the gd T cell pool of aged mice - mainly due to the selective expansion of Vg6+ gd17 T cells and augmented gd17-polarisation of Vg4+ T cells." The mechanism(s) leading to increased proportion and number of IL-17-producing gd17 T cells is unclear. "Expansion" suggests proliferation but it is not demonstrated, while "augmentation" does not distinguish between proliferation, recruitment or cell death of the other subsets. Thus, it would be appropriate to use augmentation instead of expansion. Then, it would be interesting to clarify the mechanism of augmentation of Vg6+ gd17 T cells and gd17-polarisation of Vg4+ T cells. To do so, it is recommended to perform an experiment of BrdU incorporation, or Ki67 staining and caspase 3 staining of gd17 T cells harvested from mice at 12 months and 21 months.

> We have performed the experiment and neutralized IL-7 *in vivo* in combination with EdU labelling (*new Fig. 5I*). Strikingly, we find that the proliferation of gd17 T cells, in particular Vg6+ gd17 T cells, but not gd1 T cells in the pLN is mediated by IL-7 (*new Fig. 5J and K, new Suppl. Fig. 9, and new Suppl. Fig. 10*).

12-month old mice have more Vg6+ gd17 T cells and the proportions of these cells can be more easily assessed. Although the EdU incorporation was lower compared with young mice (<10%) and we did only block IL-7 for 4 days we were able to see a slight decrease in the proportion of Vg6+ gd17 T cells.

These results are nicely in line with our previous results showing that IL-7 is highly expressed in the aged LN (*Fig. 5A and G*), confirming that gd17 T cells express higher levels of IL-7Ra compared with gd1 T cells (*Fig. 3A, Fig. 5B*), and demonstrating that Vg6+ gd17 T cells are localised in close proximity to IL-7 producing cells (*Fig. H*).

Taken together, we propose that IL-7-producing cells in the pLNs create a niche for local Vg6+ gd17 T cell expansion.

3) P7 L318 > *"Strikingly, we found that 3LL-A9 tumours developed faster in old mice (Fig. 6C)." To support that tumours grow faster in old compared to young mice, it would be informative to plot the tumour growth (X= days; Y= tumour size (mm³)) by calculating the tumour volume with H and L (as measured with a calliper). The approximation of the tumour volume is then given with the following formula: (L x H x H)/2 and plotted as a function of time (days). Visualisation of the kinetics of tumour growth will be more convincing than only one measurement given in grams 14 days after transplantation of the tumour.*

> We have added a *new panel to Fig. 6 (Fig. 6B)* showing the tumour growth kinetics over the time of the experiment in young and old animals to more convincingly show that 3LL-A9 tumours grow faster in old mice.

5) *Evidence has emerged of a crosstalk between gd17 T cells and neutrophils in various contexts (such as cancer and infections) (Cheemarla et al., 2017; Liu et al., 2016; Mensurado et al., 2018; Wozniak et al., 2012). In particular neutrophils have been shown in tumours to limit the presence of gd17 T cells in a way that affects tumour growth (Liu et al., 2016; Mensurado et al., 2018). Thus, beside a role of IL-7 on creating a favourable niche for gd17 T cells in the lymph nodes, it would be interesting to determine a potential role for neutrophils in tumours of old mice. To do so assessment of neutrophils (CD45+ live/dead-ve CD11b+ Ly6Cint Ly6GHi) by flow cytometry should be performed in tumours harvested from young and old mice.*

> We have now assessed neutrophil infiltration in tumours from young and old mice. The results are shown in the *new panel Fig. 6I*.

The proportion of CD11b+, Ly6Cint, Ly6Ghi neutrophils in total live CD45+ tumour-infiltrating immune cells was not significantly different between tumours from young and old mice, respectively, arguing against a major role of neutrophils in determining the different growth kinetics between young and old mice in the 3LL-A9 model. However, 3 of the 8 young mice analysed had slightly increased neutrophil infiltration.

6) P8 L360 > *"Furthermore, the biased gd17 T cell pool in pLNs of old mice correlates with a pro-tumour microenvironment and enhanced tumour progression." While the existence of a gd17 T cell bias with age is convincing, its physiologic relevance still remains to be clearly established. To establish clearly a correlative link between gd17 T cells and tumour progression, it would be critical to assess the tumour growth in young and old mice deficient in IL-17 (IL-17-/- mice) and gd T cells (TCRd-/- mice). As an alternative, injection of blocking antibodies specific for mouse IL-17 and TCRd (such as GL3, and GL4 (Goodman and Lefrancois, 1989; Koenecke et al., 2009; Sheridan et al., 2013)) should be considered and followed by direct measurements of tumour growth. Of note: it was shown that the GL3 antibody induces a state of anergy of the gd T cells due to TCR internalisation, instead of being depleted (Koenecke et al., 2009), nevertheless this approach should impair IL-17+ invariant Vg6 T cells that can recognise tumour-associated antigens or other signals, at least in the 3LL-A9 model (see P10 L415).*

> Physiological ageing of IL-17-/- and TCRd -/- mice is not feasible as this would entail waiting for the mice to reach 21-month of age, which is a long and costly experiment that is unlikely to yield meaningful results, because these mice are not ideally suited to firmly link gd17 T cells with tumour progression. In IL-17-/- mice, the production of IL-17 by Th17 CD4+ T cells and other cell types is also abolished; and in TCRd-/- mice, both anti-tumour gd1 and pro-tumour gd17 lineages are

eliminated. The use of blocking antibodies against TCR δ and IL-17 is associated with the same caveats.

To address the reviewer's concern and functionally link the known pro-tumour activity of gd17 T cells with enhanced tumour growth, we attempted to determine the effect of pan-TCRgd depletion on 3LL-A9 tumour growth in old mice (Figure for reviewers). Although the treatment with anti-TCRgd antibody abolished the detection of gd T cells in the tumour (panel B), functional blockade of gd T cells in old mice did not consistently abrogate the growth of 3LL-A9 tumour in aged mice over a period of 14 days. Rather we observed a heterogeneous response in which some of the tumours within UC7-13D5-treated mice grew similar to isotype control whereas others had reduced growth (panel C). The presence of neutrophils, macrophages, CD4⁺ T cells and CD8⁺ T cells in the TME was not affected by functional blockade of gd T cells (panels D-F).

As pointed out by the reviewer, the use of anti-TCR δ mAbs like GL3 and UC7-13D5 does not lead to depletion of gd T cells of either lineage (gd1 and gd17). The gd T cells internalize their T cell receptors and become undetectable but persist. Most importantly, the upregulation of activation markers on gd T cells has been reported after administration of anti-TCR δ mAbs *in vivo* (Koencke *et al.*, 2009). These properties of anti-TCR δ mAbs prevent a conclusive interpretation of the results from the above experiments, which is thus not a suitable approach to functionally interrogate the role of enriched Vg6⁺ gd T cells in promoting tumour growth in aged mice. In view of this, we have chosen not to include the data in the study but present it for the reviewers only.

Minor Issues:

1) There is a typo in figure 5E on the Y axis "% Vg6+ ceels in all T cells" should be changed to "% Vg6+ cells in all T cells".

> The typing error has been corrected.

2) Please refrain from using "tumour development" because tumours only develop in spontaneous genic mouse models of cancer (such as mice with mutations in p53 and Brca1 (breast) or Braf and Pten (skin) for example...). When tumour cells are injected subcutaneously, that is implanted, the development phase is bypassed and we should refer to tumour growth, progression or assessment of tumour burden.

> We agree with the referee and have substituted "tumour development" with "tumour growth" throughout the manuscript.

3) P7 L300> "Taken together, we show that IL-7 production in the T cell zone of pLNs is highly increased upon ageing and correlates with the expansion of gd17 T cells, especially Vg6⁺ T cells, resulting in a skewed peripheral gd T cell pool" P8 L360 > "Furthermore, the biased gd17 T cell pool in pLNs of old mice correlates with a pro-tumour microenvironment and enhanced tumour progression." As indicated above, the present work does not provide clear demonstration that IL-7 is indeed the extrinsic cytokine in the microenvironment that promotes the accumulation of gd17 T cells. In the same way, there is no evidence that the presence of gd17 T cells indeed supports tumour progression. There is at best an association, but not a correlation, thus please only use "correlation" in the few appropriate contexts throughout the manuscript.

> With the new functional data that we've generated, we can firmly say that it is IL-7 that mediates the selective expansion of gd17 T cells, in particular Vg6⁺ gd17 T cells in the pLNs. Due to the lack of suitable reagents to selectively ablate gd17 T cells, we could not provide direct functional evidence that the altered gd T cell pool in the pLN of old mice leads to enhanced tumour growth and have reworded our manuscript accordingly.

References

- Jin, C., *et al.*, Commensal Microbiota Promote Lung Cancer Development via gammadelta T Cells. Cell, 2019. 176(5): p. 998-1013 e16.
- Koencke, C., *et al.*, In vivo application of mAb directed against the gammadelta TCR does not deplete but generates "invisible" gammadelta T cells. Eur J Immunol, 2009. 39 (2): p. 372-9.
- Nakamizo, S., *et al.*, High fat diet exacerbates murine psoriatic dermatitis by increasing the number of IL-17-producing gammadelta T cells. Sci Rep, 2017. 7(1): p. 14076.
- Wilharm, A., *et al.*, Mutual interplay between IL-17-producing gammadeltaT cells and microbiota orchestrates oral mucosal homeostasis. Proc Natl Acad Sci U S A, 2019. 116(7): p. 2652-2661.

Thank you for the submission of your revised research manuscript to EMBO reports. We have now received reports from the three referees that were asked to re-assess your study, which can be found at the end of this email.

As you will see, all referees now support the publication of your manuscript in EMBO reports. The referees have some further suggestions and comments, we ask you to address in a final revised version. Please discuss the points mentioned by the referees in the final manuscript text.

When submitting your revised manuscript, please also carefully review the instructions that follow below. Failure to include requested items will delay the publication of your study. When submitting your revised manuscript, we will require:

- 1) a .docx formatted version of the final manuscript text (including legends for main figures, EV figures and tables - see below point 2), but without the figures included. Please make sure that changes made compared to the previous version are highlighted and clearly visible. Figure legends should be compiled at the end of the manuscript text.
- 2) individual production quality figure files as .eps, .tif, .jpg (one file per figure), of main figures and EV figures. Please upload these as separate, individual files.

The Expanded View format, which will be displayed in the main HTML of the paper in a collapsible format, has replaced the Supplementary information. You can submit up to 5 images as Expanded View. Please follow the nomenclature Figure EV1, Figure EV2 etc. The figure legend for these should be included in the main manuscript document file in a section called Expanded View Figure Legends after the main Figure Legends section. Additional Supplementary material should be supplied as a single pdf labeled Appendix. The Appendix should have page numbers and needs to include a table of content on the first page (with page numbers) and legends for all content. Please follow the nomenclature Appendix Figure Sx, Appendix Table Sx etc. throughout the text, and also label the figures and tables according to this nomenclature.

For more details please refer to our guide to authors:

<http://embor.embopress.org/authorguide#manuscriptpreparation>

See also our guide for figure preparation:

http://www.embopress.org/sites/default/files/EMBOPress_Figure_Guidelines_061115.pdf

- 3) a .docx formatted letter INCLUDING the reviewers' reports and your responses to their final comments. As part of the EMBO Press transparent editorial process, the point-by-point response is part of the Review Process File (RPF), which will be published alongside your paper.

- 4) Before submitting your revision, primary datasets produced in this study need to be deposited in an appropriate public database. See: <http://embor.embopress.org/authorguide#datadeposition>

The accession numbers and database should be listed in a formal "Data Availability " section (placed after Materials & Methods) that follows the model below. Please note that the Data Availability Section is restricted to new primary data that are part of this study.

Data availability

- RNA-Seq data: Gene Expression Omnibus GSE46843
(<https://www.ncbi.nlm.nih.gov/geo/query/acc.cgi?acc=GSE46843>)

- [data type]: [name of the resource] [accession number/identifier/doi] ([URL or identifiers.org/DATABASE:ACCESSION])

5) We now strongly encourage the publication of original source data with the aim of making primary data more accessible and transparent to the reader. The source data will be published in a separate source data file online along with the accepted manuscript and will be linked to the relevant figure. If you would like to use this opportunity, please submit the source data (for example scans of entire gels or blots, data points of graphs in an excel sheet, additional images, etc.) of your key experiments together with the revised manuscript. If you want to provide source data, please include size markers for scans of entire gels, label the scans with figure and panel number, and send one PDF file per figure.

6) Our journal encourages inclusion of **data citations in the reference list** to directly cite datasets that were re-used and obtained from public databases. Data citations in the article text are distinct from normal bibliographical citations and should directly link to the database records from which the data can be accessed. In the main text, data citations are formatted as follows: "Data ref: Smith et al, 2001" or "Data ref: NCBI Sequence Read Archive PRJNA342805, 2017". In the Reference list, data citations must be labeled with "[DATASET]". A data reference must provide the database name, accession number/identifiers and a resolvable link to the landing page from which the data can be accessed at the end of the reference. Further instructions are available at: <http://embor.embopress.org/authorguide#referencesformat>

7) Regarding data quantification and statistics, can you please check that, where applicable, the number "n" for how many independent experiments (biological replicates) were performed is specified, as well as the bars and error bars (e.g. SEM, SD) and the test used to calculate p-values in the respective figure legends. Please provide statistical testing where applicable. See: <http://embor.embopress.org/authorguide#statisticalanalysis>

8) Please provide the abstract written in present tense.

9) John Marioni is missing from the author contributions. Please add him and specify his contribution.

10) Please add a conflict of interest statement below the author contributions.

11) Please add up to five key words to the title page.

12) Per journal policy, we do not allow 'data not shown' (see page 9 of your manuscript, the legend of Fig. 4 and the Suppl. Fig 1). All data referred to in the paper should be displayed in the main or Expanded View figures, or the Appendix. Thus, please add these data (or change the text accordingly, if these data are not important). See: <http://embor.embopress.org/authorguide#unpublisheddata>

13) There is a call-out for Fig. 1H, but there is no such panel. There is no call-out for Fig. 1E. Please check.

14) It seems there are no separate call-outs for Suppl. Figs. 3,4,5,8 and 16. Please call these out in the text (but see point 2 above, first).

Finally, I would need from you:

- a short, two-sentence summary of the manuscript
- two to three bullet points highlighting the key findings of your study
- a schematic summary figure (in jpeg or tiff format with the exact width of 550 pixels and a height of not more than 400 pixels) that can be used as a visual synopsis on our website.

I look forward to seeing a revised version of your manuscript when it is ready. Please let me know if you have questions or comments regarding the revision.

REFEREE REPORTS**Referee #1:**

De la Roche and colleagues chose to rebut the initial rejection and substantially revised version of their ms on IL-17-producing Vg6+ T cells in ageing mice. They have adequately addressed the reviewers' previous concerns. Importantly, they now include data from interventional blocking of IL-7 with anti-IL-7 neutralizing antibody, which suggest that expansion of the gd17 T cells, in particular Vg6+ T cells, in the pLN is driven by IL-7.

This raises the question whether LN stroma of aged mice produces more IL-7 or whether there are less CD127+ IL-7-consuming in this setting, thereby favoring Vg6+ T cell expansion.

Referee #2:

The authors have substantially revised their manuscript to satisfy nearly every comment from myself and the other reviewers. It is an interesting and thorough piece of work. They have included the functional data on the IL-7 - gd17 link to support their conclusions. Now the manuscript is more than just descriptive, it is a mechanistic paper. The microbiome data is an interesting sidenote, but I do not think these data were acceptable to ask for or necessary to include. Regarding the gdT cell depletion studies, the authors may have had better luck with the GL3 clone instead of UC7-13D5; however, I don't believe these experiments were necessary for this study either. They are right to exclude the data from the final version, as the data only confound the issue. I was intrigued to see that gdT17 cells incorporate more EdU, because these cells do not usually proliferate in non-lymphoid organs. This makes me think that the cells entering/exiting lymph nodes may be a separate or specialized population.

Referee #3:

The authors have addressed (or attempted to) most of my comments, which were to provide direct in vivo functional demonstration that i) IL-7 supports a niche rich in IL-17+ Vg4 and Vg6 gd T cells in peripheral lymph nodes, and ii) that IL-17+ Vg4 and Vg6 gd T cells have a major impact on tumour growth.

They have provided evidence that IL-7 supports accumulation of IL-17-producing gd T cells by using a blocking anti-mouse IL-7 antibody. I feel the authors have missed an opportunity to fully address the two issues at once. Since there are limitations to the selective ablation of gd17 T cells with anti-TCRd Ab, the authors could have taken advantage of the effect of the neutralisation of IL-7 that leads to a decrease in IL-17-producing Vg4 and Vg6 gd T cells, without affecting CD4 and CD8 T cells. This provides a suitable way to manipulate the selectively gd17 T cells and not the gd1 T cells, and the authors could have use this approach to look at the effect of selective manipulation of gd17 T cells on tumour growth, or, at the very least commented on why not to use this approach.

2nd Revision - authors' response

29 May 2019

Referee #1:

De la Roche and colleagues chose to rebut the initial rejection and substantially revised version of their ms on IL-17-producing Vg6+ T cells in ageing mice. They have adequately addressed the reviewers' previous concerns. Importantly, they now include data from interventional blocking of IL-7 with anti-IL-7 neutralizing antibody, which suggest that expansion of the gd17 T cells, in particular Vg6+ T cells, in the pLN is driven by IL-7.

This raises the question whether LN stroma of aged mice produces more IL-7 or whether there are less CD127+ IL-7-consuming in this setting, thereby favoring Vg6+ T cell expansion.

> We have added a sentence in the manuscript (page 11, line 485) to discuss the two possible mechanisms leading to increased levels of IL-7 in the aged pLN pointed out by the referee.

Referee #2:

The authors have substantially revised their manuscript to satisfy nearly every comment from myself and the other reviewers. It is an interesting and thorough piece of work. They have included the functional data on the IL-7 - gd17 link to support their conclusions. Now the manuscript is more than just descriptive, it is a mechanistic paper. The microbiome data is an interesting sidenote, but I do not think these data were acceptable to ask for or necessary to include. Regarding the gdT cell depletion studies, the authors may have had better luck with the GL3 clone instead of UC7-13D5; however, I don't believe these experiments were necessary for this study either. They are right to exclude the data from the final version, as the data only confound the issue. I was intrigued to see that gdT17 cells incorporate more EdU, because these cells do not usually proliferate in non-lymphoid organs. This makes me think that the cells entering/exiting lymph nodes may be a separate or specialized population.

> Conceptionally it is an interesting question whether the lymph node-resident and/or lymph node-recirculating $\gamma\delta 17$ T pool represents a specialised $\gamma\delta$ T cell population with unique proliferative and activation features that are different from other peripheral $\gamma\delta 17$ T cell subsets. We've added a sentence in the manuscript to highlight this (page 11, line 503).

Referee #3:

The authors have addressed (or attempted to) most of my comments, which were to provide direct in vivo functional demonstration that i) IL-7 supports a niche rich in IL-17+ Vg4 and Vg6 gd T cells in peripheral lymph nodes, and ii) that IL-17+ Vg4 and Vg6 gd T cells have a major impact on tumour growth.

They have provided evidence that IL-7 supports accumulation of IL-17-producing gd T cells by using a blocking anti-mouse IL-7 antibody. I feel the authors have missed an opportunity to fully address the two issues at once. Since there are limitations to the selective ablation of gd17 T cells with anti-TCRd Ab, the authors could have taken advantage of the effect of the neutralisation of IL-7 that leads to a decrease in IL-17-producing Vg4 and Vg6 gd T cells, without affecting CD4 and CD8 T cells. This provides a suitable way to manipulate the selectively gd17 T cells and not the gd1 T cells, and the authors could have use this approach to look at the effect of selective manipulation of gd17 T cells on tumour growth, or, at the very least commented on why not to use this approach.

> In principle we agree with the referee. Our finding that short-term IL-7 neutralisation leads to selective depletion of $\gamma\delta 17$ T cells could potentially provide a unique opportunity to interrogate the functional role of enriched LN-resident $\gamma\delta 17$ T cells in promoting tumour growth in aged mice. For the following reasons we decided not to pursue this avenue.

First, we have shown that proliferation rates of $\gamma\delta 17$ T cells are low in the pLNs of older mice. Thus, old mice would have to be treated with IL-7 neutralizing antibodies for a long time to sufficiently ablate $\gamma\delta 17$ T cells in the pLN. Second, long-term IL-7 depletion will also severely affect the proliferation and maintenance of naïve and memory CD4⁺ and CD8⁺ T cells leading to an altered T cell repertoire upon tumour challenge. Thirdly, IL-7 is important for the anti-tumour activity of intra-tumoural CD4⁺ and CD8⁺ T cells (Andersson *et al.*, 2009). Because of these limitations we consider this approach not feasible to assess the effect of $\gamma\delta 17$ T cells on tumour growth.

Reference

Andersson A., Yang S.-K., Huang M., *et al.*, IL-7 promotes CXCR3 ligand-dependent T cell antitumor reactivity in lung cancer. *J Immunol* 2009; 182: 6951-6958.

Corresponding Author Name: Maïke de la Roche

Manuscript Number: EMBOR-2018-47379V2-Q